# Quantifying Language Models' Sensitivity to Spurious Features in Prompt Design *or:*
## *How I learned to start worrying about prompt formatting*

**Melanie Sclar**[1]    **Yejin Choi**[1,2]    **Yulia Tsvetkov**[1]    **Alane Suhr**[3]
[1]Paul G. Allen School of Computer Science & Engineering, University of Washington
[2]Allen Institute for Artificial Intelligence    [3]University of California, Berkeley
`msclar@cs.washington.edu`

### ABSTRACT

As large language models (LLMs) are adopted as a fundamental component of language technologies, it is crucial to accurately characterize their performance. Because choices in prompt design can strongly influence model behavior, this design process is critical in effectively using any modern pre-trained generative language model. In this work, we focus on LLM sensitivity to a quintessential class of meaning-preserving design choices: prompt formatting. We find that several widely used open-source LLMs are extremely sensitive to subtle changes in prompt formatting in few-shot settings, with performance differences of up to 76 accuracy points when evaluated using LLaMA-2-13B. Sensitivity remains even when increasing model size, the number of few-shot examples, or performing instruction tuning. Our analysis suggests that work evaluating LLMs with prompting-based methods would benefit from reporting a range of performance across plausible prompt formats, instead of the currently-standard practice of reporting performance on a single format. We also show that format performance only weakly correlates between models, which puts into question the methodological validity of comparing models with an arbitrarily chosen, fixed prompt format. To facilitate systematic analysis we propose FORMATSPREAD, an algorithm that rapidly evaluates a sampled set of plausible prompt formats for a given task, and reports the interval of expected performance without accessing model weights[1]. Furthermore, we present a suite of analyses that characterize the nature of this sensitivity, including exploring the influence of particular atomic perturbations and the internal representation of particular formats.

## 1 INTRODUCTION

As the capabilities of LLMs have rapidly improved, their sensitivity to input prompt features has been used to optimize performance via prompt engineering (White et al., 2023). However, there has been little work in characterizing this sensitivity, especially to seemingly innocuous feature choices that preserve prompt meaning and intent. In this work, we analyze the sensitivity of widely used, open-source LLMs to a class of features that should not influence a prompt's interpretation: formatting choices. We find that pre-trained LLMs are sensitive to these choices in unpredictable ways, with accuracy varying in up to 76 points for LLaMA-2-13B between equivalent formats, and ∼10 accuracy points on average across 50+ tasks and several models. We also show that this variance is not eliminated by adding few-shot examples, increasing model size, or instruction tuning.

Designing prompt templates is a critical part of effectively using a pre-trained language model. This design process includes making choices about wording, choosing few-shot examples for in-context learning, and making decisions about seemingly trivial features like formatting. This process, and often even the resulting templates, is rarely reported or discussed in research papers, under the assumption that performance variance across these choices is insignificant compared to variance across data points or models. However, some anecdotal evidence points to formatting choices actually having a significant influence on model behavior (Aghajanyan, 2023). In some cases, researchers report a limited number of manually generated formats to show that scaling trends hold despite perfor-

---

[1]FORMATSPREAD's code can be found at `https://github.com/msclar/formatspread`.

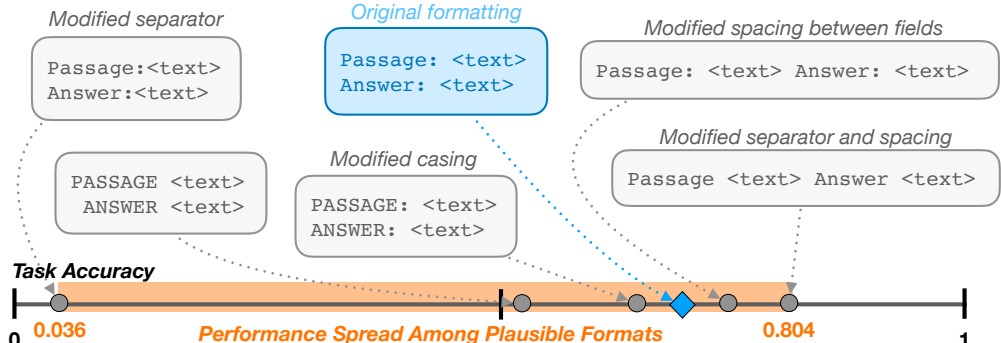

Figure 1: Slight modifications in prompt format templating may lead to significantly different model performance for a given task. Each `<text>` represents a different variable-length placeholder to be replaced with actual data samples. Example shown corresponds to 1-shot LLaMA-2-7B performances for task280 from SuperNaturalInstructions (Wang et al., 2022). This StereoSet-inspired task (Nadeem et al., 2021) requires the model to, given a short passage, classify it into one of four types of stereotype or anti-stereotype (gender, profession, race, and religion).

mance being significantly different (Schick et al., 2021). The assumption that formatting does not influence overall model performance may become problematic when improvements over existing approaches are attributed to the amount and source of training data, number of parameters, or model architecture, without also accounting for changes in prompt format. Ignoring variance across formats may also negatively affect user experience, e.g. if users inadvertently choose formats the LLM does not perform well on.

Our proposed tool, FORMATSPREAD, enables a systematic analysis of these variances across a wide set of semantically equivalent prompt formats within a user-specified computational budget. We find that choices in formatting few-shot examples during in-context learning introduce spurious biases that may lead to significantly different conclusions in model performance. The sensitivity to formatting choices that we discover across widely-used, open-source models suggests that future research would benefit from reporting a performance *spread* over a sufficient sample of plausible formats, instead of simply reporting the formatting used and its performance, as is currently standard. Moreover, we argue that this reporting is crucial when comparing the performance of different models, as we show the influence of formatting choices only weakly correlates between models, thus making and fixing a formatting choice could introduce a significant confounding factor.

Fully exploring the space of prompt formats is intractable, as computation costs scale linearly with the number of formats considered. FORMATSPREAD efficiently explores the space of prompt formats under a user-specified computational budget using Bayesian optimization. FORMATSPREAD does not require access to the model weights, allowing its use on API-gated models: we find a spread up to 56 accuracy points with a median spread of 6.4 accuracy points with GPT3.5 across 320 formats and 53 tasks at a cost of under 10USD on average per task. Beyond facilitating evaluation, we also propose a suite of analyses to further characterize model sensitivity to formatting. Among other results, we show that the separability of continuous prompt embeddings correlates with the spread observed in task performance.

## 2 OVERVIEW

We evaluate LLM performance over the space of prompt formats that may plausibly be chosen by a non-adversarial user when designing a prompt for a target task, where the space of formats is defined by a grammar (§3.1). Our grammar's definition naturally induces a definition of semantic equivalence among formats. We quantify model sensitivity in terms of performance range in a target task across the space of equivalent prompt formats to the original choice (§4.2). We cast the problem of searching across this space as a bandit problem, and propose FORMATSPREAD (§3), which consists of a grammar (§3.1) and a procedure to estimate the minimum and maximum performance across a set of semantically equivalent formats given a pre-defined metric (§3.2). FORMATSPREAD uses Bayesian optimization to identify the expected performance range with low additional computational cost (§4.5) all without requiring access to model weights, which enables use on API-gated

LLMs. Furthermore, we perform in-depth analysis of this observed sensitivity, including by quantifying the contribution of individual feature choices to the final performance (§4.3) and measuring the identifiability of a format based solely on a model's internal, continuous representation of any prompt via correlation with model performance (§4.4).

## 3 MEASURING SENSITIVITY WITH FORMATSPREAD

### 3.1 GRAMMAR OF PLAUSIBLE PROMPT FORMATS

We construct a grammar that defines both the space of plausible prompt formats and semantic equivalence between formats. The grammar is manually constructed, as opposed to automatically induced from data, to guarantee a higher level of precision when defining the set of equivalent formats. Our grammar is directly tested by verifying that it can generate the formatting associated with 100+ Super-NaturalInstructions tasks (Wang et al., 2022).

Our grammar consists of fields that are composed to create a prompt format. For example, the format 'Passage: <text> || Answer: <text>', has basic fields 'Passage: <text>', and 'Answer: <text>', denoted $a_1$, and $a_2$. Each basic field consists of a *descriptor* (e.g. 'Passage'), a *separator* (e.g. ': '), and a text placeholder to replace with each data point. We define basic fields as $B_1(d, s, f) := f(d)s\texttt{<text>}$ using Backus-Naur notation, where $d$ is a descriptor string, $s \in \mathcal{S}_1$ a separator, and $f \in \mathcal{F}_{\text{casing}}$ a function that alters $d$ while preserving meaning. Thus, in our example, $a_1 = B_1(\texttt{Passage}, ': ', id)$ and $a_2 = B_1(\texttt{Answer}, ': ', id)$, with $id$ the identity function. We define joining several fields as $B_2^{(n)}(X_1, \ldots, X_n, c) := X_1 c X_2 c \ldots c X_n$, with $c \in \mathcal{C}$ being a *space*. Our example's prompt format may be written as $B_2^{(2)}(a_1, a_2, ' || ')$.

The grammar also supports enumeration, which is defined as joining several basic fields, each representing a different list item. For example, the enumeration 'Option (A): <text>, Option (B): <text>, Option (C): <text>' may be written as $B_2^{(3)}(a_1, a_2, a_3, ' || ')$, where $a_i = B_1(e_i, ': ', id)$. In our example, $e_1$ represents 'Option (A)', and may in turn be written as the concatenation $e_i := ds_2 f_{\text{item}}(i)$ with $d = $ 'Option', $s_2 = ' '$ (single space), and $f_{\text{item}}(1) = $ '(A)'. Each $f_{\text{item}}$ transforms an item $i$ using a number format (e.g. letters or Roman numerals, denoted as $\mathcal{F}_{\text{item2}}$) and an item wrapper (e.g. (A) or [A], denoted $\mathcal{F}_{\text{item1}}$).

In summary, we define valid prompt formats as those accepted by the following grammar:

$$B_0() := \texttt{<text>}$$
$$B_0'(d, s) := f(d)s \quad \text{with } s \in \mathcal{S}_1, \ f \in \mathcal{F}_{\text{casing}}$$
$$B_1(d, s, f) := f(d)s\texttt{<text>} \quad \text{with } s \in \mathcal{S}_1, \ f \in \mathcal{F}_{\text{casing}}$$
$$B_2^{(n)}(X_1, \ldots, X_n, c) := X_1 c \ldots c X_n \quad \text{with } c \in \mathcal{C}, \ X_i \in \{B_0, B_0', B_1, B_2, B_3\} \ \forall i$$
$$B_3^{(n)}(d, j_1, \ldots, j_n, s_1, s_2, c, f_1, f_2) := B_2^{(n)}(B_1(e_1, s_1, f_2)), \ldots, B_1(e_n, s_1, f_2), c)$$
$$\text{where } e_i := f_2(d) \ s_2 \ f_1(j_i), j_i \in \mathbb{N}_0 \ \forall i,$$
$$s_1 \in \mathcal{S}_1, s_2 \in \mathcal{S}_2, f_1 \in \mathcal{F}_{\text{item}}, f_2 \in \mathcal{F}_{\text{casing}}$$

Our grammar defines valid formats as finite compositions of $B_0, B_0', B_1, B_2, B_3$. The sets $\mathcal{S}_1, \mathcal{S}_2, \mathcal{C}$, $\mathcal{F}_{\text{casing}}, \mathcal{F}_{\text{item}}$ (two sets of separators, spaces, casing functions, and itemizing functions respectively) are pre-defined by the user. Throughout this work, we instantiate all sets with values typically observed in human-written prompt formats. We intentionally only modify the casing of descriptors (via $\mathcal{F}_{\text{casing}}$) to guarantee semantic equivalence; one may also define a set of functions that paraphrases the descriptor, e.g., via synonym replacement. Appendix A.2 contains the full list of values we use for the constant sets, as well as a visualization of a prompt template generated from the grammar.

**Prompt Format Equivalence.** Two prompt formats $p_1, p_2$ are equivalent if they represent the same rule application $B_i$, the descriptors (if any) are the same, and the sub-elements (if any) are equivalent. Appendix A.1 contains the formal definition of equivalence. The grammar's strict definition allows us to assume that sets of equivalent formats share equivalent meanings. When measuring sensitivity (§3.2), we explore only the space of formats equivalent to a task's original format.

**Contextual Restrictions.** We define restrictions to the combinations of spaces and separators to further ensure naturalness. For example, if $B_2(X_1, \ldots, X_n, c)$ where $c$ does not contain a newline, then each $X_i$'s separators and any subcomponents' separators should not contain a newline. This

avoids unnatural formats like `Input:\n <text> Output:\n <text>`. We also allow for adding conditions that force constants (separators, spaces, etc.) in different applications of $B_i$ to be equal. When measuring sensitivity to format perturbations, if two separators or spaces are equal in an original format, they are forced to jointly change to be considered equivalent. Appendix A.3 contains all contextual restrictions.

**Final Prompt Construction.** Given a valid format $p$ accepted by the grammar, the final prompt is constructed by concatenating with space $c$ an instruction string $inst$, $n$ few-shot data points $D_1, \ldots, D_n$ exemplifying the task, and a data point $D_{n+1}$ to be solved. All few-shot examples $D_i$ are formatted using $p$. Thus, the final prompt template is: $inst \; c \; p(D_1) \; c \; p(D_2) \; c \; \ldots \; c \; p(D_n) \; c \; p(D_{n+1})$. Since $D_{n+1}$'s output will be generated by the model, an empty string is added in place of the answer in the last field in the template. Prompt construction will modify $inst$ to match specific choices encoded in $p$: concretely, if $p$ enumerates valid multiple-choice options as characters $x_1 \ldots x_n$, we ensure $inst$ refers to these choices as $x_1 \ldots x_n$.

## 3.2 MEASURING SENSITIVITY

We measure how plausible choices in prompt formatting influence quantifiable metrics of generated outputs. Given a set of plausible formats $\{p_1, \ldots, p_n\}$, a dataset $\mathcal{D}$, and a scalar metric $m$, let the *performance interval* be $[\min_i m(p_i, \mathcal{D}), \max_i m(p_i, \mathcal{D})]$. We define the *performance spread* or simply *spread* as $\max_i m(p_i, \mathcal{D}) - \min_i m(p_i, \mathcal{D})$. Higher spread indicates more sensitivity to variance within the space of plausible, semantically-equivalent formats. While our method is agnostic to the scalar metric $m$ used, and one could consider a number of metrics including text length, formality, or toxicity, throughout this work we focus our analysis on estimated task accuracy $acc$. Due to ease in automatic evaluation, here we evaluate on classification tasks.

Our goal is to compute spread for a given model and task. A comprehensive approach would be to fully evaluate each plausible format $p_i$ on the entire evaluation dataset $\mathcal{D}$. This increases the cost of reporting a model's performance linearly with $n$, which becomes computationally infeasible for large values of $n$. Following prior gradient-free prompt engineering work (Zhou et al., 2023; Pryzant et al., 2023), we model our problem as a multi-arm bandit. Given a random sample of $n$ formats (arms) $p_1, \ldots, p_n$ for a task, an arm $p_i$'s hidden value is the actual performance $m(p_i, \mathcal{D})$ when evaluated on the full dataset $\mathcal{D}$, and the reward for pulling the arm is an estimate $m(p_i, \tilde{\mathcal{D}})$ where $\tilde{\mathcal{D}} \subset \mathcal{D}, |\tilde{\mathcal{D}}| = B$ (mini-batch size) and no element of $\tilde{\mathcal{D}}$ has yet been evaluated with $p_i$.

We assume a budget of $E$ total data point evaluations. We first search for the highest performing format with budget $E/2$, and then for the lowest performing format with budget $E/2$. Evaluations done for the first exploration are readily available for the second exploration, which yields a more informative prior for many formats. We consider two well-known regret minimization bandit algorithms: Thompson sampling (used in FORMATSPREAD) and Upper Confidence Bound (UCB).

**Thompson Sampling.** This simple, high-performing Bayesian inference heuristic randomly draws each arm according to its probability of being optimal (Chapelle & Li, 2011). Each $m(p_i, \mathcal{D})$ is modeled as a random variable, and since with our target metric each data point evaluation is a Bernoulli trial, it is natural to model $m(p_i, \mathcal{D})$ as a Beta distribution. In each round, Thompson sampling draws from each $m(p_i, \tilde{\mathcal{D}})$ and chooses the best arm $\hat{i}$ (Algorithm 1). It then updates $\hat{i}$ according to the number of observed successes $r$, and the corresponding $B - r$ failures, within $\tilde{\mathcal{D}}$.

---

**Algorithm 1** Thompson Sampling for Bernoulli Bandits

---

$S_i^{(1)} \leftarrow 0, N_i^{(1)} \leftarrow 0$ (success counters and total times armed was drawn counter)
**for** $t \leftarrow 1, \ldots E/B$ **do**
    **for** $i \leftarrow 1, \ldots, K$ **do**
        Take $\theta_i^{(t)}$ from $\text{Beta}(\alpha_i + S_i^{(t)}, \beta_i + (N_i^{(t)} - S_i^{(t)}))$
    Draw arm $\hat{i} = \arg\max_i \theta_i^{(t)}$ (or $\arg\min$ in minimization problems) and observe reward $r$
    $S_{\hat{i}}^{(t+1)} \leftarrow S_{\hat{i}}^{(t)} + r, \quad N_{\hat{i}}^{(t+1)} \leftarrow N_{\hat{i}}^{(t)} + B$

---

Thompson sampling allows for setting informative priors $(\alpha_i, \beta_i)$ based on domain knowledge to accelerate runtime. Appendix A.4 details the exact priors we use. To our knowledge, we are the first to consider a Bayesian sampling method for prompt optimization.

**Upper Confidence Bound (UCB) Sampling.** UCB (Lai et al., 1985) computes an upper confidence bound to each arm's performance, derived from Chernoff's bound. The key difference with Thompson sampling is in how $\theta_i^{(t)}$ is defined. In UCB's frequentist approach, $\theta_i^{(t)}$ is assigned the estimated accuracy plus the upper confidence bound: $\theta_i^{(t)} \leftarrow S_i/N_i + c\sqrt{log(t)/N_i}$. We use $c = 2$ following Pryzant et al. (2023), who find UCB with $c = 2$ to be most effective for prompt optimization.

**Naive Sampling.** Each prompt format is evaluated on $E/n$ points (with appropriate rounding).

# 4    CHARACTERIZING PROMPT FORMAT VARIANCE WITH FORMATSPREAD

## 4.1    EXPERIMENTAL SETUP

**Data.** We use a subset of 53 tasks from Super-NaturalInstructions (Wang et al., 2022) with diverse human-written formats and instructions, comprising 19 multiple-choice tasks and 34 classification tasks with $\{2, 3, 4\}$ basic fields. Appendix B.1 details the exact task selection procedure. To construct the final prompt template, we concatenate each task's instruction and $n$ formatted few-shot examples using \n\n as spacing. While selection and ordering of few-shot examples is a component of prompt design influencing features of model output (Lu et al., 2022), our work focuses on prompt formatting. To remove this confounder, we fix the exact choice and ordering of examples for each task and for a given number of shots $n$. Few-shot examples for each task are chosen randomly within each dataset and are not used for evaluation. We evaluate task data samples on an arbitrary order fixed across settings. Datasets are assumed to be of size 1,000 for fair evaluation across tasks.

**Models.** We evaluate LLaMA-2-$\{7B,13B,70B\}$ (Touvron et al., 2023), Falcon-7B and Falcon-7B-Instruct (Almazrouei et al., 2023), GPT-3.5-Turbo (Schulman et al., 2022), all autoregressive LMs.

**Task Evaluation Metrics.** We use two popular measures for computing accuracy: exact prefix matching and probability ranking. In exact prefix matching, we check if the output's prefix matches the expected answer after normalization (casing, spacing, newlines). Ranking accuracy computes the rate that the expected answer is the highest-ranked valid option (in multiple choice and classification tasks) according to the model's output distribution. Results are reported using ranking accuracy unless specified otherwise. Appendix B.2 shows additional analysis of exact prefix matching, with spreads even higher than those shown in Section 4.2, and including how formatting choice affects task degeneration (i.e., not answering any valid option).

## 4.2    PROMPT FORMATS HAVE A LARGE PERFORMANCE SPREAD, NOT ELIMINATED BY INCREASING FEW-SHOT EXAMPLES OR MODEL SIZE, NOR WITH INSTRUCTION TUNING

For each evaluation task we randomly sample 10 plausible prompt formats and use FORMATSPREAD to compute performance spread for each modeling and $n$-shot choice (Figure 3). We find significant performance spread across tasks, with a median spread of 7.5 accuracy points across choices in the model and the number of few-shot examples. 20% of tasks consistently result in a spread of at least 15 accuracy points for all LLaMA-2 settings, and at least 9 points for all Falcon settings. We observe several tasks with performance spread over 70 accuracy points. Because this analysis uses only 10 randomly sampled formats, it represents a lower bound of the true spreads for each task. Furthermore, there exists significant performance spread regardless of increased model size (Figure 2a and Figure 11 for Llama-2-70B), instruction tuning (Figure 2b), or number of few-shot examples (Figure 2c; Figure 2a and 2b plot 1- and 5-shot jointly). Appendix B.2 demonstrates similar results on a selection of non-classification tasks, and expands the spread discussion to plotting the entire accuracy distribution, along with a dispersion metric.

**Comparison trends between models are often reversed just by choosing different formats.** Assuming model $M$ is better than $M'$ by at least $d$ accuracy using prompt $p$, we compute how often $M'$ achieves at least $d$ higher accuracy than $M$ under a different format $p'$. Figure 4 shows these trends are often reversed: LLaMA-2-13B and -70B reverse trend by at least $d = 0.02$ with probability 0.141; LLaMA-2-7B and Falcon-7B reverse trend by at least $d = 0.02$ with probability 0.140. Strikingly, often both experiments (first using $p$, and then $p'$) were statistically significant (p-value $< 0.05$) on 1000 samples[2]: 76% and 47% respectively for the two model comparisons

---

[2]We use one-sided McNemar tests, also known as paired $\chi^2$ tests, since we evaluate models on the same set of samples. We test the significance of $M$ being *better* than $M'$ under $p$, and $M$ being *worse* than $M'$ under $p'$.

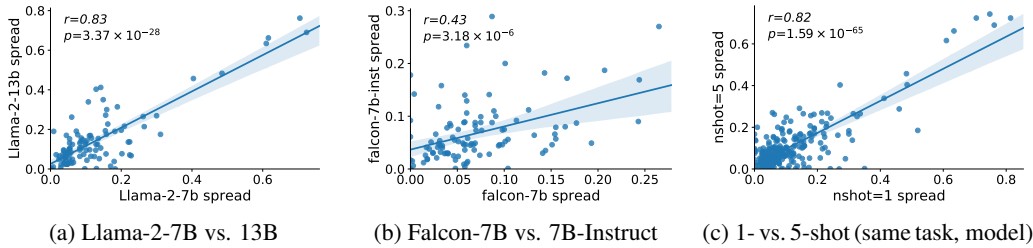

(a) Llama-2-7B vs. 13B      (b) Falcon-7B vs. 7B-Instruct      (c) 1- vs. 5-shot (same task, model)

Figure 2: Spread comparison between evaluating the same task under different models or $n$-shots.

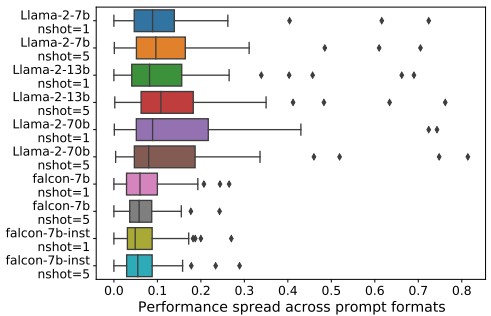

Figure 3: Spread across models and $n$-shots.

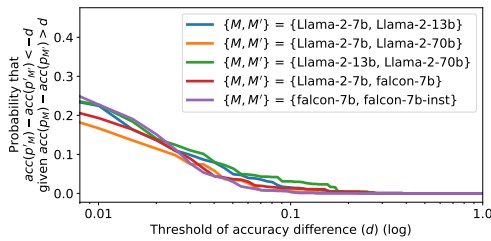

Figure 4: Probability that model $M$ performs *worse* than $M'$ by at least $d$ when using format $p'$, given that $M$ performed *better* than $M'$ by at least $d$ using format $p$. 53 tasks, 1- and 5-shot.

mentioned above. We find that formats yielding high performance for model $M$ may not yield high performance for $M'$, implying that **formats may not be inherently good or bad** (Appendix B.2).

### 4.3 HOW DO INDIVIDUAL FEATURES CONTRIBUTE TO PERFORMANCE?

We analyze how choices in particular constants (i.e. $\mathcal{S}_1, \mathcal{S}_2, \mathcal{C}, \mathcal{F}_{\text{casing}}, \mathcal{F}_{\text{item}}$) independently influence task performance across different formats. Figure 5 shows the distribution of accuracy for 500 sampled prompts conditioned on the choice of $\mathcal{S}_1$ (the separator between a descriptor and the text placeholder) for one task in Super-NaturalInstructions. When comparing the individual influence of two feature choices, we measure both *weak* and *strong* notions of dissimilarity between distributions of accuracy across prompts conditioned on a chosen feature. We say two constant choices yield *weakly* different accuracy distributions if the values between the first quartile ($Q_1$) and third quartile ($Q_3$) do not intersect. This is equivalent to the boxes in a boxplot not overlapping. We say two constant choices yield *strongly* different accuracy distributions if the ranges $[2.5Q_1 - 1.5Q_3, 2.5Q_3 + 1.5Q_1]$ do not overlap (adjusted to end in a data point). This is equivalent to two boxplots with their whiskers not overlapping. In Figure 5, '$\backslash$n$\backslash$t' and ':   ' (fourth and sixth) are only weakly different.

We compute accuracy for 500 random formats with 250 samples each on 31 tasks for 1-shot Llama-2-7B. Table 1 shows that choices in $\mathcal{S}_2, \mathcal{F}_{\text{item1}}, \mathcal{F}_{\text{casing}}$ do not independently predict performance differences (weakly or strongly): although these features can have a large performance variance and thus should be explored with FORMATSPREAD, they cannot be used to independently predict accuracy changes. Other constant sets have varying degrees of differences, with $\mathcal{S}_1$ (separators) and $\mathcal{F}_{\text{item2}}$ (number format changes in enumerations) having the most individual impact. All tasks with strong dissimilarities are shown in Appendix B.4.

**Small prompt variations often yield large performance differences.** Table 2 shows a selection of tasks where changing a single constant on a format (e.g., casing in task322) results in large accuracy differences. Figure 6 shows that regardless of the scoring criterion used, a significant ratio of these atomic changes are associated with large accuracy changes. For example, 24% of atomic changes have an associated accuracy change of at least 5 points when using exact prefix matching as scoring criteria (11% when using probability ranking).

The space of prompt format accuracy is highly non-monotonic, which makes local search algorithms over the space less effective. Let $(p_1, p_2, p_3)$ be a prompt format triple such that $p_{i+1}$ is obtained by making an atomic change to $p_i$. We argue that if the prompt format space is smooth, we should often see a triples' accuracy to be strictly monotonic over $i$. We choose 24 tasks (13 multiple choice,

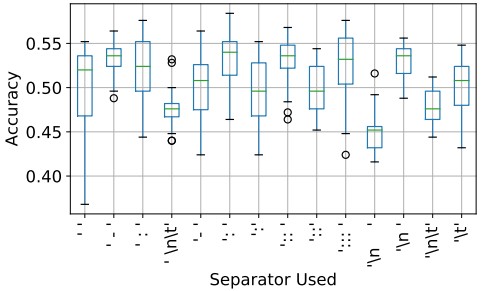

Figure 5: Example of accuracy variance for different choices of constants in $\mathcal{S}_1$ for task1283.

Table 1: Tasks where at least two constants yield different performance (*weakly* different if their boxes in a boxplot do not overlap, *strongly* if boxes including whiskers do not overlap).

|  | Median Spread (range [0, 1]) | Weak Diff. | Strong Diff. |
|---|---|---|---|
| $\mathcal{C}$ | 0.144 | 29% | 1% |
| $\mathcal{S}_1$ | 0.132 | 43% | 22% |
| $\mathcal{S}_2$ | 0.238 | 0% | 0% |
| $\mathcal{F}_{\text{item1}}$ | 0.176 | 0% | 0% |
| $\mathcal{F}_{\text{item2}}$ | 0.173 | 45% | 10% |
| $\mathcal{F}_{\text{casing}}$ | 0.188 | 3% | 0% |

Table 2: Examples of atomic changes' impact on accuracy using probability ranking (prefix matching shown in Table 4). {} represents a text field; $p_2$ yields higher accuracy than $p_1$ for all tasks.

| Task Id | Prompt Format 1 ($p_1$) | Prompt Format 2 ($p_2$) | Acc $p_1$ | Acc $p_2$ | Diff. |
|---|---|---|---|---|---|
| task280 | `passage:{}\n answer:{}` | `passage {}\n answer {}` | 0.043 | 0.826 | 0.783 |
| task317 | `Passage::{} Answer::{}` | `Passage:: {} Answer:: {}` | 0.076 | 0.638 | 0.562 |
| task190 | `Sentence[I]- {}Sentence[II]- {}` `-- Answer\t{}` | `Sentence[A]- {}Sentence[B]- {}` `-- Answer\t{}` | 0.360 | 0.614 | 0.254 |
| task904 | `input:: {} \n output:: {}` | `input::{} \n output::{}` | 0.418 | 0.616 | 0.198 |
| task320 | `target - {} \n{} \nanswer - {}` | `target - {}; \n{}; \nanswer - {}` | 0.361 | 0.476 | 0.115 |
| task322 | `COMMENT: {} ANSWER: {}` | `comment: {} answer: {}` | 0.614 | 0.714 | 0.100 |
| task279 | `Passage : {}. Answer : {}` | `PASSAGE : {}. ANSWER : {}` | 0.372 | 0.441 | 0.069 |

11 non-multiple choice), sample 300 $(p_1, p_2, p_3)$ triples for each, and the compute accuracy (using exact prefix matching) of each $p_i$ on 250 samples. 32.4 and 33.6% of triples were monotonic for multiple-choice and non-multiple-choice tasks respectively. Given that random shuffling within a triple will result in monotonicity 33.3% of the time, this suggests that local search mechanisms like simulated annealing may not be effective as they require a locally smooth search space.

## 4.4 PROMPT FORMATS ARE IDENTIFIABLE TRANSFORMATIONS OF PROMPT EMBEDDINGS

Prompt format choices represent a deterministic transformation of the input, even if its impact on the resulting performance is hard to predict. We represent prompt embeddings as the last hidden layer obtained when processing the whole input prompt (immediately before generating the first token). We demonstrate that format choice yields a highly identifiable transformation over this embedding, which suggests that formats can be seen as transformations of the output probability distribution.

For each task, and for both {1, 5}-shot settings, we collect prompt embeddings from LLaMA-2-7B corresponding to 10 randomly sampled valid formats for 1000 evaluation examples. We train an XGBoost (Chen & Guestrin, 2016) classifier that maps from the top $n$ principal components of a prompt embedding to the prompt format.[3] We find that although the original prompt embeddings are of size 4,096[4], using just the top 100 principal components can result in a classifier with $\geq 0.98$ accuracy in format identification for all 31 tasks analyzed. Figure 7 shows the accuracy of format classification given a fixed number of principal components.[5] We find that classifier accuracy given just the top two components correlates moderately with the spread of performance in the prompts they represent ($0.424, p = 8.04 \cdot 10^{-6}$; 0.555 for the 5-shot setting; using exact prefix matching).

## 4.5 FAST EXPLORATION OF THE PROMPT FORMATTING SPACE: FORMATSPREAD

In Section 4.2, we demonstrate that even when sampling just 10 formats from the space of plausible formats, we still observe significant performance spread on many tasks. However, this is only a lower

---

[3]We train with 800 vectors from each of the 10 formats (8000 vectors) and evaluate on the remaining 200.

[4]Equivalent to the dimension of hidden representations for LLaMA-2-7B.

[5]Figure 21 in the Appendix visualizes examples of the top two principal components for ten prompt formats.

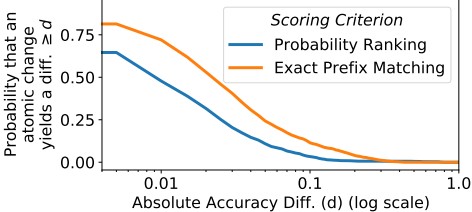

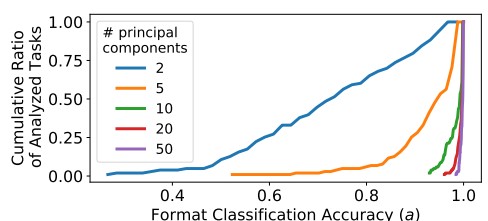

Figure 6: Probability that an atomic change (e.g. changing a space, separator) has a given impact in accuracy for two scoring criteria. 53 tasks, 30 sampled atomic changes each.

Figure 7: Cumulative ratio of tasks that can be classified with at most $a$ accuracy using the top principal components of the last decoding layer of the prompt.

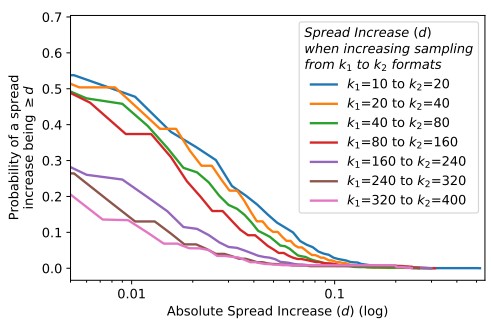

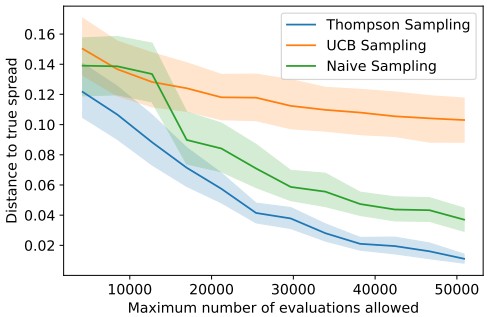

Figure 8: Probability of observing a spread increase of at least $d$ when increasing sample size from $k_1$ to $k_2$ formats. 31 tasks, 100 trials each.

Figure 9: Difference between the true sample spread and each algorithm-found spread with respect to $E$ (evaluation budget). 320 formats, $B = 20$, average of 5 trials over 31 tasks shown.

bound of the spread a task may exhibit when increasing the number of formats: for example, about 17% of tasks are expected to increase their spread by at least 5 accuracy points when increasing from 10 to 20 sampled formats. Figure 8 quantifies the expected increase in spread when increasing the number of formats by evaluating 500 formats on 250 samples each and computing expected gains.

Figure 9 compares the efficiency of Thompson sampling, UCB, and naive sampling for estimating spread with respect to a budget $E$ (Section 3.2). To ensure accurate reports, we compute and show the true spread of the highest- and lowest-performing formats chosen by each method using all data. With a budget of 51,200 evaluations, Thompson sampling results in a spread within 1 accuracy point of the true spread, while naive sampling finds a spread within 4 points, and UCB within 11.

Finally, we use FORMATSPREAD to measure sensitivity of several models where inference is expensive. With a budget of 40,000 evaluations and 320 prompt formats, we find that 1-shot LLaMA-2-70B–ran using 4-bit quantization (Dettmers et al., 2022)–yields a median spread of 0.171 (mean=0.221, std=0.200, using probability ranking across 53 tasks; 25% of tasks had a spread of 0.292 or higher, with a maximum spread of 0.876), and GPT-3.5 yields a median spread of 0.064 (mean=0.110, std=0.115, across 53 tasks using exact prefix matching given that we do not have access to the full logits; 25% of tasks had a spread of 0.148 or higher, with a maximum spread of 0.562), showing sensitivity to formatting is still present even on larger models. 5-shot LLaMA-2-70B still shows high spreads, with 25% of tasks having a spread of 0.310 and a maximum of 0.841. See spread visualization in Figure 25, and a list of best and worst formats found in Table 6.

## 5 RELATED WORK

The task of automatically finding the best-performing prompt for a given task without changing model parameters has recently gained attention, given the constantly improving yet somewhat unpredictable performance of LLMs. Prior work has often focused on discovering optimal prompts with gradient-based methods, which are effective, but often lead to disfluent or unnatural prompts (Shin

et al., 2020), which can be mitigated with a Langevin dynamics-based method (Shi et al., 2022). Another approach is to learn, optimize, and insert continuous representations of prompts and tasks as input to models (Qin & Eisner, 2021; Lester et al., 2021; Ding et al., 2022; Ilharco et al., 2023). These methods also require access to the LLM's parameters, thus cannot be applied to models behind an API. In contrast, FORMATSPREAD does not assume access to any model internals. Prior gradient-free work has focused on edit-based enumeration over human-written prompts (Prasad et al., 2023), reinforcement learning (Deng et al., 2022), and by using LLMs themselves (Zhou et al., 2023; Gao et al., 2021). These works aim to achieve competitive task performance, even if the meaning of the prompt or instruction is modified. To our knowledge, we are the first to focus specifically on prompt formatting variance, a quintessential example of semantic equivalence.

Jailbreaking refers to the behavior of intentionally manipulating prompts to elicit inappropriate or sensitive responses, or otherwise reveal parts of the prompt that were intentionally not revealed. While the objective differs from our work, jailbreaking works (Wei et al., 2023; Zou et al., 2023) share the underlying technical question of finding the lowest-performing prompt. Our methods differ, since Wei et al. (2023) evaluate human-generated attacks to guide adversarial prompt design, and Zou et al. (2023) uses gradient-based search methods simultaneously across multiple models.

Some existing work has explored the influence of certain prompt design choices on model performance, for example the prompt's language (Gonen et al., 2022), the ordering of few-shot examples (Lu et al., 2022), and their patterns (Madaan et al., 2023). Other work has focused on providing textual interpretations of continuous prompt representations (Khashabi et al., 2022). Beyond autoregressive LLMs, existing work has focused on performance variance in masked language models (Elazar et al., 2021; Jiang et al., 2020). Our work follows efforts in other domains that explore the influence of spurious features on research evaluations, e.g., in deep reinforcement learning (Islam et al., 2017; Henderson et al., 2018) and statistical machine translation (Clark et al., 2011).

## 6 DISCUSSION

We introduce FORMATSPREAD, an algorithm that estimates the performance *spread* across prompt formatting choices.[6] We use FORMATSPREAD to evaluate the spread of several widely-used open-source LLMs for classification tasks in few-shot learning settings. We find that spread is large regardless of model choice, even when increasing model size, number of few-shots, or when using instruction tuning. FORMATSPREAD is designed to efficiently search the space of plausible prompt formats under a user-specified computational budget. For example, with a computational budget of exploring only 5% of the entire search space for task with 2,500 test examples and 320 plausible formats, we are able to estimate spread within 2 accuracy points of the true spread.

We also characterize the space of prompt formats, finding that it is largely non-monotonic and that few atomic features can be predictors of performance alone, although the separability of format embeddings is highly correlated with observed performance spread. These findings informed the design of our search procedure, where local search methods are not advantageous.

Our findings suggest that performance spread caused by arbitrary prompt formatting choices may influence conclusions made about model performance, especially when comparing models on benchmark tasks. Thus, we recommend that work evaluating LLMs with prompting-based methods would benefit from reporting a range of performance across plausible formats. However, we want to emphasize that single-format evaluation may still be sufficient for many use cases. For example, for researchers or practitioners who build systems on top of LLMs, choosing a single prompt format that works sufficiently well for use in this larger system is a valid methodological choice. However, we encourage future research to compute FORMATSPREAD when comparing their systems to out-of-the-box models, to ensure fair baseline representation. Furthermore, FORMATSPREAD can be used to identify lower-bound performance of a model or system. For example, when using a model for socially impactful tasks, such as stereotype classification in Figure 1, it is important to report the range of accuracy a non-adversarial user might encounter. Likewise, it is crucial to consider robustness to spurious features when claiming that models possess general abilities, such as theory of mind; and beneficial to report when e.g. exploring model biases. We leave it to future research to develop regularization procedures either during training or with an already-trained model to make models robust to diverse formatting choices.

---

[6]We thoroughly describe the limitations of our method in Appendix C.

## 7 ACKNOWLEDGEMENTS

We thank Jillian Fisher, Sachin Kumar, Angela Zhou, and the Berkeley NLP group for valuable discussions. This work was conducted while A.S. was a Young Investigator at AI2. This material is based upon work partly funded by the DARPA CMO under Contract No. HR001120C0124, by DARPA MCS program through NIWC Pacific (N66001-19-2-4031), by NSF DMS-2134012, IIS-2125201, IIS-2203097, by NSF CAREER Grant No. IIS2142739, and an Alfred P. Sloan Foundation Fellowship. Any opinions, findings and conclusions or recommendations expressed in this material are those of the authors and do not necessarily state or reflect those of the United States Government or any agency thereof.

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

## A    GRAMMAR DEFINITION AND INSTANTIATION DETAILS

### A.1    EQUIVALENCE RELATION DEFINITION

Precisely, $p_1 \sim p_2$ if and only if at least one of the following hold: $p_1 = p_2 = B_0$; or $p_i = B_0'(d_i, s_i)$ with $d_1 = d_2$; or $p_i = B_1(d_i, s_i, f_i)$ with $d_1 = d_2$; or $p_i = B_2^{(n)}(X_{1,i}, \ldots, X_{n,i}, c_i)$ with $X_{j,1} \sim X_{j,2} \ \forall 1 \le j \le n$; or $p_i = B_3^{(n)}(d_i, j_{1,i}, \ldots, j_{n,i}, s_1, s_2, c, f)$ where $d_1 = d_2$ and $j_{k,1} = j_{k,2} \ \forall 1 \le k \le n$. It is possible that generated formats equivalent in their string representation are not equivalent according to this equivalence relation.

### A.1.1    VISUALIZATION OF PROMPT FORMAT'S PARSING AND FULL FORMAT GENERATION

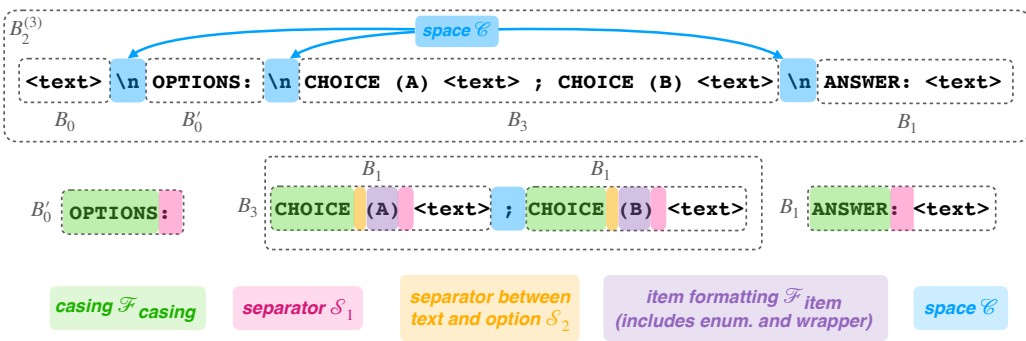

Figure 10: Visualization of a complex prompt format showing its parsing and which constants or functions affect each part of the format.

Figure 10 shows a visualization of how a complex format is parsed using our defined grammar. A full prompt consists of an instruction, n few-shots and a data point to solve. For example, if the instruction was `Given a sentence and two words that appear in it, answer which one of the two (A or B) appeared first in the sentence.`, a full prompt may look as follow. Note that we always use \n\n as space character between instruction and few-shots. The example below shows a 1-shot prompt. It is simply illustrative and does not correspond to any of the tasks considered.

```
Given a sentence and two words that appear in it, answer which one of
the two (A or B) appeared first in the sentence.

The quick brown fox jumps
OPTIONS:
CHOICE (A): fox ; CHOICE (B): brown
ANSWER: B

Over the lazy dog
OPTIONS:
CHOICE (A): lazy ; CHOICE (B): dog
ANSWER:
```

FORMATSPREAD forces all instantiations of a multiple choice variable to change jointly to maintain coherence, and this includes text in the instruction. Therefore, when changing the option items from A and B to I and II, the prompt will be generated as follows.

```
Given a sentence and two words that appear in it, answer which one of
the two (I or II) appeared first in the sentence.

The quick brown fox jumps
OPTIONS:
CHOICE (I): fox ; CHOICE (II): brown
ANSWER: II

Over the lazy dog
OPTIONS:
CHOICE (I): lazy ; CHOICE (II): dog
ANSWER:
```

## A.2 ALLOWED VALUES FOR EACH SET $\mathcal{S}_1, \mathcal{S}_2, \mathcal{C}, \mathcal{F}_{\text{CASING}}, \mathcal{F}_{\text{ITEM}}$

$$\mathcal{S}_1 = \{'', ' ', ' \backslash n', ' \backslash n', ' - - ', ' ', '; \backslash n', ' \| ', ' < \text{sep} > ', ' - - ', ', ', ' \backslash n', ' , ', ' \backslash n', '. ', ' , '\}$$
$$\mathcal{S}_2 = \{'', ' ', '  ', ' \backslash t'\} \text{ (no space, single space, double space, tab)}$$
$$\mathcal{C} = \{'', '::: ', ':: ', ': ', ' \backslash n \backslash t', ' \backslash n ', ' : ', ' - ', ' ', ' \backslash n ', ' \backslash n \backslash t', ':', ':: ', ' - ', ' \backslash t'\}$$
$$\mathcal{F}_{\text{casing}} = \{\mathbf{f(x) = x}, \mathbf{f(x) = x.title()}, \mathbf{f(x) = x.upper()}, \mathbf{f(x) = x.lower()}\}$$
$$\mathcal{F}_{\text{item}} = \{x \mapsto f(g(x)) \mid \text{such that } f \in \mathcal{F}_{\text{item1}} \wedge g \in \mathcal{F}_{\text{item2}}\}$$
$$\mathcal{F}_{\text{item1}} = \{\mathbf{x \mapsto (x)}, \mathbf{x \mapsto x.}, \mathbf{x \mapsto x)}, \mathbf{x \mapsto x_{\sqcup}}, \mathbf{x \mapsto [x]}, \mathbf{x \mapsto < x >}\}$$
$$\mathcal{F}_{\text{item2}} = \{\mathbf{x \to x + 1}, \mathbf{x \to' A' + x}, \mathbf{x \to' a' + x},$$
$$\mathbf{x \to 0x215F + x + 1}, \mathbf{x \to ROMAN[x].lower()}, \mathbf{x \to ROMAN[x].upper()}\}$$

Enumerations are indexed from (i.e., "1, 2, 3" rather than "0, 1, 2"). $\texttt{ROMAN}[\texttt{x}]$ represents the Roman numerals written in regular ASCII characters. `0x215F`+x represent the series of Unicode characters for Roman numerals. $\sqcup$ denotes a spacing character for clarity.

## A.3 RESTRICTIONS TO PROMPT FORMATS SPACES AND SEPARATORS' COMBINATIONS

We define several restrictions to ensure format naturalness. Users can additionally customize FORMATSPREAD by defining their own rules and restrictions between values. Our rules are as follows:

- If $B_2(X_1, \ldots, X_n, c)$ where $c$ does not contain a newline, then each $X_i$'s separators and any subcomponents' separators should not contain a newline.

- Similar to the rule above, if $B_3^{(n)}(d, j_1, \ldots, j_n, s_1, s_2, c, f_1, f_2)$ such that some separator contains a newline (i.e. $s_1$ contains a newline and/or $s_2$ contains a newline) then the space $c$ must also contain a newline.

- For $B_1(d, s, f) := f(d)s< \texttt{text} >$, $s$ must not be the empty string (i.e., there has to be some separation between descriptor and text).

- Having $c$ be an empty string space in $B_2^{(n)}$ is only allowed if the first $n - 1$ components are $B_1$ fields with an empty $<\texttt{text}>$. Similarly, the newline restrictions mentioned above only apply if the $<\texttt{text}>$ is not empty. This rarely happens in prompt formats, but there are formats such as $\texttt{Question:}$ $<\texttt{text}>$ $\texttt{Options:}$ $\texttt{A. <text> B. <text>}$ where the $\texttt{Options:}$ do not have a corresponding field.

## A.4 THOMPSON SAMPLING PRIORS

For the first exploration (i.e., finding the best-performing prompt format), we set an informative prior $\text{Beta}(\alpha, \beta) := \text{Beta}\left(\max\left(\frac{\beta \cdot x}{1-x}, 1.1\right), 5\right)$ for all arms $p_i$, where $x$ is the original format's accuracy. Our goal is to set an informative prior where the expected value of the prior distribution is the original format accuracy $x$, since a priori it is the only information we have about performance.

This restricts the parameters as follows:

$$\mathbb{E}[\text{Beta}(\alpha, \beta)] = \frac{\alpha}{\alpha + \beta} = x$$

$$\alpha = \alpha \cdot x + \beta \cdot x$$

$$\alpha = \frac{\beta \cdot x}{1 - x}$$

Since $\beta$ will modulate how confident is the prior, and we want to avoid the model being overconfident, we fix $\beta = 5$. Because we want to have an informative prior $\text{Beta}(\alpha, \beta)$ with a Gaussian-like PDF, we force $\alpha > 1$ and $\beta > 1$. In extreme cases, forcing $\alpha > 1$ might alter the expected value. The first exploration's priors are thus exactly $\text{Beta}(\alpha, \beta)$ with $\alpha = \max\left(\frac{\beta \cdot x}{1-x}, 1.1\right)$ and $\beta = 5$ for all arms $p_i$.

For the second exploration (i.e., finding the worst-performing prompt format), the model has access to the first explorations' counters $S_i^{(E/B)}$ and $N_i^{(E/B)}$. Therefore, we set the second exploration's priors to be $\text{Beta}\left(\alpha + S_i^{(E/B)}, \beta + \left(N_i^{(E/B)} - S_i^{(E/B)}\right)\right)$.

## B  ADDITIONAL EXPERIMENTS' INFORMATION AND PLOTS

### B.1  TASK SELECTION

We use a number of heuristics to filter Super-NaturalInstructions tasks to our set of 53 evaluation tasks. Datasets should have at least 1000 samples to be considered. We also remove tasks whose instructions are too long (over 3,000 characters) and datasets with inputs longer than 2,000 characters, given that this makes performing inference at scale intractable. We also filter datasets whose valid outputs include more than 20 different strings, given that we focus on classification tasks.

We also removed tasks where we found a priori performance on the task was 0% accuracy using LLaMA-2-7B 1-shot. Some Super-NaturalInstructions tasks are derived from the same original dataset, but ask different questions. We did not include more than 4 tasks from the same original dataset.

Finally, we also searched for having socially impactful tasks. Those tasks were the only Super-NaturalInstructions tasks where we included a format if one was not provided by the dataset.

The selected tasks were the following 53: `task050, task065, task069, task070, task114, task133, task155, task158, task161, task162, task163, task190, task213, task214, task220, task279, task280, task286, task296, task297, task316, task317, task319, task320, task322, task323, task325, task326, task327, task328, task335, task337, task385, task580, task607, task608, task609, task904, task905, task1186, task1283, task1284, task1297, task1347, task1387, task1419, task1420, task1421, task1423, task1502, task1612, task1678, task1724`.

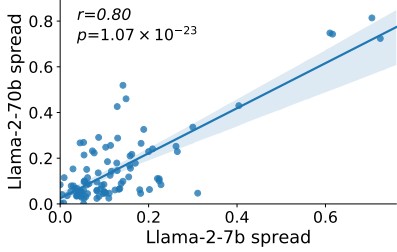

Figure 11: Comparison between Llama-2-7B and Llama-2-70B spreads. Llama-2-70B was computed using 4bit quantization (Dettmers et al., 2022).

## B.2 ADDITIONAL RESULTS FOR SECTION 4.2

Table 3: Ratio of prompt format pairs $(p_1, p_2)$ such that if $p_1$ is worse than $p_2$ using model $M_1$, then the same trend holds for $M_2$.

| Model 1 $(M_1)$ | Model 2 $(M_2)$ | Performance Relative Ordering Preservation |
|---|---|---|
| Llama-2-7b | Llama-2-13b | 57.46% |
| Llama-2-7b | Falcon-2-7b | 55.91% |
| Falcon-7b | Falcon-7b-Inst | 61.11% |

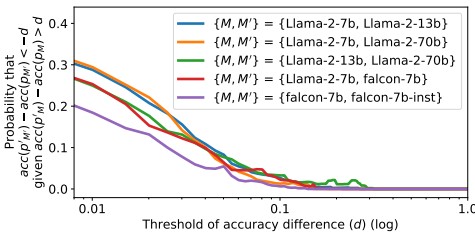

(a) Option ranking metric.

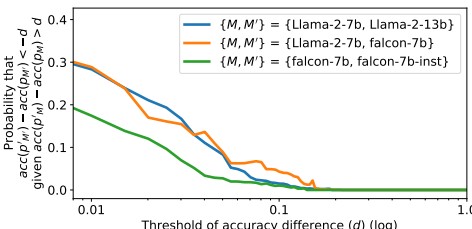

(b) Exact prefix matching metric.

Figure 12: Probability of a prompt $p$ being worse than $p'$ by at least $d$ points using model $M'$, given that prompt $p$ was *better* than prompt $p'$ when using model $M$.

**Formats are not inherently good or bad.** Table 3 shows that if format $p_1$ has lower performance than format $p_2$ under model $M$, there is $< 0.62$ probability that this trend would hold under another model $M'$ (random chance is $0.5$). This weak relative order preservation suggests that prompt format performance in a model may not be extrapolated to a different model, or in other words, that there are no inherently good or bad formats. This finding is further supported by Figure 12, which shows that findings of a format being better or worse than another are often inconsistent across models.

**Experiments with exact prefix matching accuracy.** Here we show results with using exact prefix matching to compute accuracy. Often, failures in prefix matching are associated with degeneration, i.e., cases where the model does not answer any of the valid options, motivating the use of ranking accuracy. Degeneration makes models (specially smaller models) more unlikely to have high accuracy out of the box. As seen in Figure 6, prefix matching is linked to having higher changes when performing atomic changes. Moreover, exact prefix matching can lead to lower performance as generation is less constrained (see Figure 16). Table 4 shows examples of atomic changes yielding large accuracy changes with exact prefix matching metric.

Figure 13c shows spread remains regardless of model size increase, architecture change, or number of few-shot examples also when using exact prefix matching as accuracy metric. In line with the results shown for probability ranking in Section 4.2, Figure 15 shows that the probability of reversing performance trends between two models just by changing prompt remains high when using exact prefix matching as metric. Strikingly, spread is significantly higher than in the probability ranking setting (see Figure 14), with median spread ranging from 12 to 28 accuracy points depending on the model used. This further motivates the need for running FORMATSPREAD when benchmarking models with this accuracy metric. This increased spread may be partly due to degeneration, as we will detail next.

**Degeneration.** Sometimes when a model does not generate the correct answer with exact prefix matching, it also does not generate a valid response, i.e. it degenerates. We will now quantify this phenomenon using 53 SuperNaturalInstructions classification and multiple choice tasks.

Given a model, a task, and a format, let the *centered mass* be the ratio of examples where the model's output matched with any valid option (regardless of correctness). Table 5 shows that the correlation between accuracy and centered mass is moderate or high depending on the model. This suggests that very often when a model does not return a valid answer, it does not return any valid answer

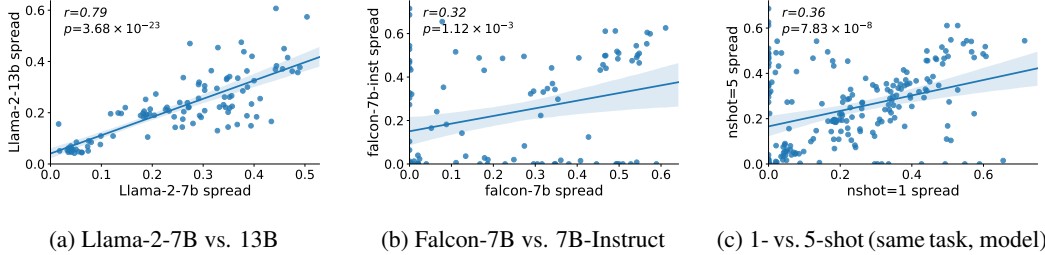

(a) Llama-2-7B vs. 13B    (b) Falcon-7B vs. 7B-Instruct    (c) 1- vs. 5-shot (same task, model)

Figure 13: Spread comparison between evaluating the same task under different models or n-shots using exact prefix matching as accuracy metric.

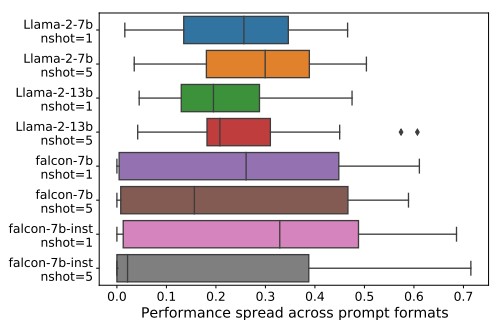

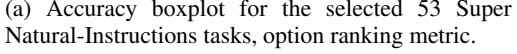

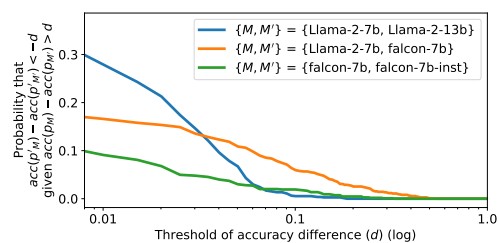

Figure 15: Probability that model $M$ performs *worse* than $M'$ by at least $d$ when using format $p'$, given that $M$ performed *better* than $M'$ by at least $d$ using format $p$. 53 tasks, 1- and 5-shot. Exact prefix matching metric.

Figure 14: Spread across models and $n$-shots. Exact prefix matching metric.

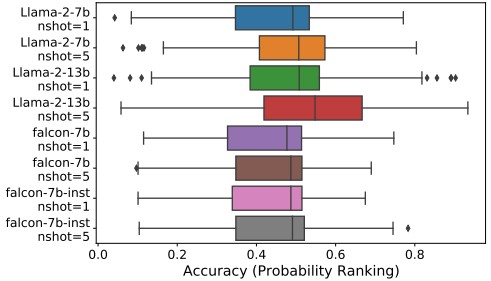

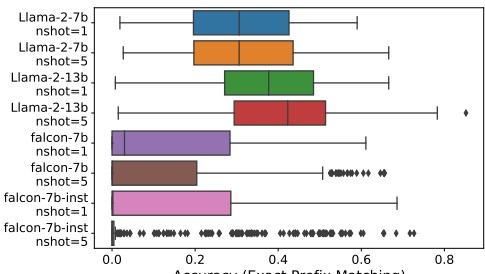

(a) Accuracy boxplot for the selected 53 Super Natural-Instructions tasks, option ranking metric.

(b) Accuracy boxplot selected 53 Super Natural-Instructions tasks, exact prefix matching metric.

Figure 16: Accuracy metric used can strongly impact final performance. 53 Super Natural-Instructions tasks shown. Ranking accuracy yields higher accuracies overall.

at all. This is especially true for Falcon models, where we observe an almost perfect correlation between accuracy and centered mass. In conclusion, prompt format chosen often do not solely affect accuracy, but they also affect the frequency in which a model is actually able to perform a task. This will especially affect tasks for which there are no alternative metrics. Further research may focus specifically on targeting features that cause degeneration.

**Experiments with Instruction Induction tasks.** All experiments thus far focused solely on classification tasks. We will now focus on tasks that require generating (short) text, and cannot be framed as classification tasks. We selected 10 tasks from Instruction Induction (Honovich et al., 2023) that require generating a unique, valid string to be considered a correct response. Examples include identifying the second letter of a word, adding numbers, or answering a synonym to a given word. Instruction Induction tasks also show a wide range of difficulty, resulting in varied settings to be

Table 4: Examples of atomic changes' impact on accuracy using prefix matching (probability ranking shown in Table 2). {} represents a text field; $p_2$ yields higher accuracy than $p_1$ for all tasks.

| Task Id | Prompt Format 1 ($p_1$) | Prompt Format 2 ($p_2$) | Acc $p_1$ | Acc $p_2$ | Diff. |
|---------|-------------------------|-------------------------|-----------|-----------|-------|
| task213 | `Title: {} Sentence<A>: {} || Sentence: {} || Sentence<C>: {} || Sentence<D>: {} Choices: \n ::: {} \n <ii>::: {} Answer: {}'` | `Title::{} Sentence<A>::{} || Sentence::{} || Sentence<C>::{} || Sentence<D>::{} Choices::\n ::: {} \n <ii>::: {} Answer::{}'` | 0.113 | 0.475 | 0.362 |
| task296 | `Sentence I ) : {} \nSentence II ) : {} \nSentence III ) : {} \nSentence IV ) : {} \nSentence V ) : {} \nSentence VI ) : {} \nSentence VII ) : {} \nSentence VIII ) : {} \nSentence IX ) : {} \nSentence X ) : {} , I. : {} , II. : {} , Answer: {}'` | `Sentence I ) : {} \nSentence II ) : {} \nSentence III ) : {} \nSentence IV ) : {} \nSentence V ) : {} \nSentence VI ) : {} \nSentence VII ) : {} \nSentence VIII ) : {} \nSentence IX ) : {} \nSentence X ) : {} , 1. : {} , 2. : {} , Answer: {}'` | 0.201 | 0.522 | 0.321 |
| task905 | `Tweet::: {}; \nLabel::: {}; \nAnswer::: {}'` | `Tweet::{}; \nLabel::{}; \nAnswer::{}'` | 0.252 | 0.559 | 0.307 |
| task317 | `Passage:: {} \nAnswer:: {}'` | `Passage::{} \nAnswer::{}'` | 0.245 | 0.546 | 0.301 |
| task280 | `passage {}\n answer {}'` | `passage:{}\n answer:{}'` | 0.332 | 0.612 | 0.28 |
| task050 | `SENTENCE - {} \nQUESTION - {} \nANSWER - {}'` | `SENTENCE\n\t{} \nQUESTION\n\t{} \nANSWER\n\t{}'` | 0.244 | 0.504 | 0.26 |
| task070 | `Beginning - {}\nMiddle [I]{} , Middle [II]{}\nEnding - {}\nAnswer - {}'` | `Beginning - {}\nMiddle I){} , Middle II){}\nEnding - {}\nAnswer - {}'` | 0.143 | 0.3 | 0.157 |

Table 5: Correlation between accuracy using exact prefix matching and the centered mass (the opposite of degeneration). 53 tasks, 10 formats each, evaluated on 1000 samples.

| Model | n-shot | correlation between accuracy & and centered mass | p-value |
|-------|--------|--------------------------------------------------|---------|
| Llama-2-7b | 1 | 0.702 | 5.1E-77 |
| Llama-2-7b | 5 | 0.762 | 4.9E-98 |
| Llama-2-13b | 1 | 0.639 | 5.8E-61 |
| Llama-2-13b | 5 | 0.662 | 9.2E-67 |
| falcon-7b | 1 | 0.936 | 7.1E-233 |
| falcon-7b | 5 | 0.933 | 8.4E-228 |
| falcon-7b-instruct | 1 | 0.962 | 3.6E-289 |
| falcon-7b-instruct | 5 | 0.958 | 5.5E-277 |

analyzed (see Figure 18b). Given that the collection does not contain human-generated formats, we applied a simple '`Input: {}\n Output: {}`' format. Results for 1-shot and 5-shot settings show spread is still high across models and n-shot choices (see Figure 17).

Tasks are: `antonyms`, `diff`, `first_word_letter`, `larger_animal`, `letters_list`, `num_to_verbal`, `second_word_letter`, `singular_to_plural`, `sum`, `synonyms`.

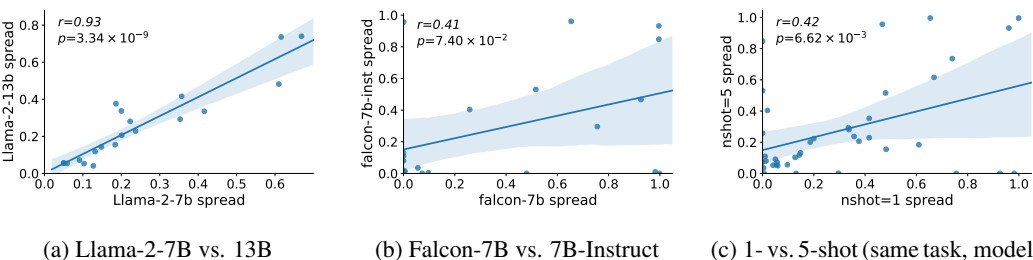

(a) Llama-2-7B vs. 13B    (b) Falcon-7B vs. 7B-Instruct    (c) 1- vs. 5-shot (same task, model)

Figure 17: Spread comparison between evaluating the same task under different models or n-shots for Instruction Induction tasks. Exact prefix matching used as accuracy metric.

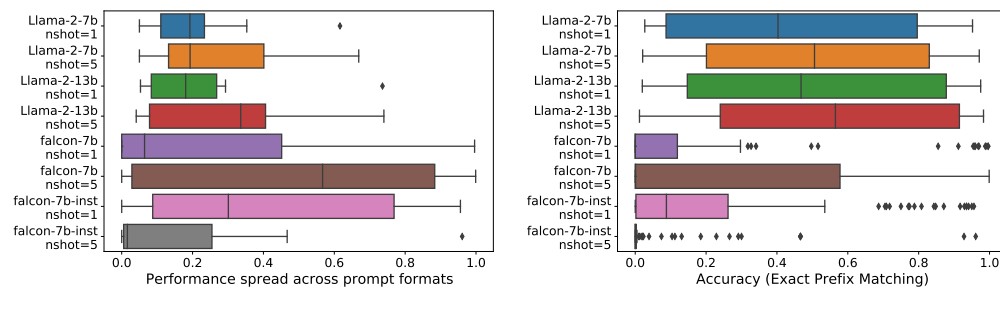

(a) Spreads across models and $n$-shots.

(b) Accuracy variance by model

Figure 18: Instruction Induction tasks' spreads and accuracy across models. Exact prefix matching is used as accuracy metric.

**Experiments with continuous metrics in open-ended text generation tasks.** Throughout the paper we focus on tasks with a single valid output, whether in classification tasks or in short-text generation tasks. This decision is intentional, since it guarantees that a variation in the metric truly represents a variation in model performance. We have shown that spread remains high when considering option ranking or exact prefix matching as accuracy metric.

Since LLMs are often used in more open-ended generation contexts, we will now explore the performance variance across prompt formats when considering sentence-length generation tasks (e.g. generate the next sentence of a story, given the four initial sentences of a story, generate a question whose answer is the sentence given). To analyze the automatic generations, we use two widely used metrics: ROUGE-L (Lin, 2004), and BERTScore (Zhang et al., 2019). The first is an n-gram-based metric, and the latter is a model-based metric, and both are $[0, 1]$ metrics where higher is better. Figure 19 shows that variance remains high for LLaMA-2-7B regardless of the metric and the number of n-shots considered, with LLaMA-2-7B 5-shot having 25% of tasks with a ROUGE-L spread of 0.098 or higher, and a BERTScore spread of 0.09 or higher.

We observe that the median spread is sometimes smaller than in the accuracy tasks. This may be because although ROUGE, BERTScore, and accuracy are all $[0, 1]$ metrics, typical metric values may be different, which may in turn affect the final spread (an absolute difference). We leave it to future work to quantify the differences in style or content that each format may be inducing.

Finally, it is worth noting that text generation metrics are known to be noisier, and thus not all metric decreases necessarily correspond to a true performance loss, as is the case for accuracy in single-valid-output tasks. We used 17 SuperNatural Instructions tasks: `task037`, `task038`,

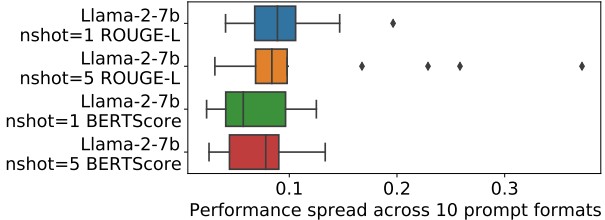

Figure 19: Spread across n-shots for LLaMA-2-7B, considering ROUGE-L and BERTScore metrics. 17 sentence-level open-generation tasks are considered, all extracted from SuperNatural Instructions. 10 prompt formats are considered for each task.

`task040`, `task067`, `task071`, `task072`, `task105`, `task216`, `task223`, `task240`, `task348`, `task389`, `task443`, `task845`, `task1326`, `task1401`, `task1613`. We selected the 17 open-ended text generation tasks among those with at least 1000 samples, with some formatting present in the original task (e.g. `'Passage:' <text>`). We only considered tasks whose instructions were under 1,000 characters and that contained inputs no longer than 5,000 characters.

We limit generations to 50 tokens. To parse model outputs more faithfully, and given that none of our expected generations include a newline, we only consider a model's generation up to the first newline (excluding leading spaces and newlines in a given generation). This consideration is important given that often models start to generate a new data sample from scratch, immediately after generating the requested answer.

**Characterizing a model's accuracy distribution beyond spread.** Spread gives a quantitative jump in information with respect to informing a single point in the performance distribution since it measures the distribution range (maximum minus minimum). However, distributions that may share the same range, may yield a widely different probability of obtaining each value in the distribution. Figure 20 plots the accuracy distribution of 30 tasks, sorted in decreasing order by standard deviation. Tasks with high standard deviation reflect a higher likelihood of obtaining dissimilar values when making a formatting selection; Figure 20 shows that the median standard distribution is $\sigma \approx 0.04$, which can be considered high in our context.

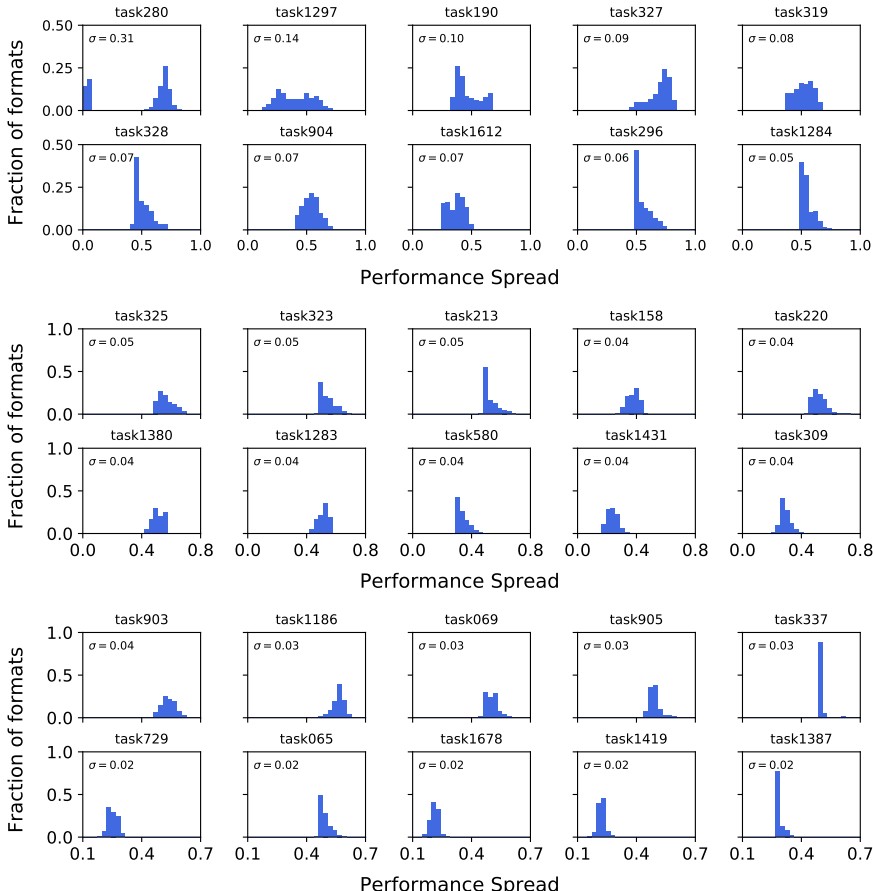

Figure 20: Accuracy distribution across 500 formats for 30 tasks evaluated on 250 samples each, sorted by standard deviation in decreasing order. LLaMA-2-7B 1-shot, option ranking metric.

**On factors influencing spread besides prompt formatting.** We believe many factors beyond formatting may be influencing performance variance, but were unable to find a feature that reliably predicts spread. We found that the average prompt length in a task has a negligible correlation with its performance spread: $r = 0.228$ ($p = 1.4 \times 10^{-7}$) for exact prefix matching metric, and $r = -0.022$ ($p = 0.615$) for option ranking metric, when jointly considering all models and n-shots. Similarly, the standard deviation of the prompt length had negligible correlation with spread: $r = 0.125$ ($p = 0.004$) for exact prefix matching, and $r = -0.099$ ($p = 0.024$) for option ranking metric. When considering each model individually, only LLaMA-2-7B with exact prefix matching showed a correlation $|r| > 0.5$, with the average prompt length having a correlation $r = 0.559$ $p = 6.86 \times 10^{-10}$. All other settings had $|r| < 0.36$.

## B.3  PCA EXAMPLES

Section 4.4 systematically analyzes whether we can predict the prompt format that generated a given pre-softmax activation layer (i.e., prompt embeddings) by using solely its top-$n$ principal components. Figure 21 shows the top two principal components for two different tasks where all 10 formats considered are easily identifiable solely with a prompt embedding's top two principal compoenents.

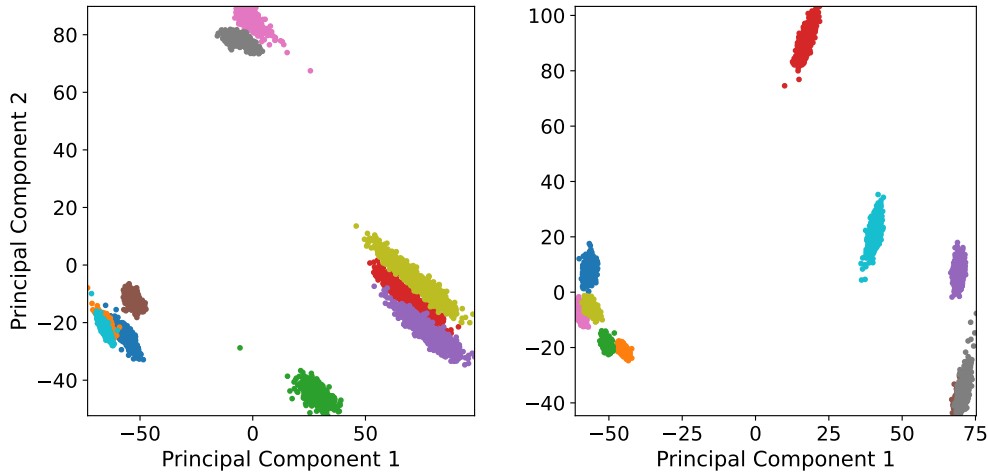

Figure 21: Plot of the top two principal components of the last decoder layer of the prompt, as a representation of the output probability distribution. Two different tasks shown, with each prompt format shown in a different color.

## B.4  NOTABLE FEATURES

As discussed in Section 4.3, sometimes the choice of a constant may lead to significantly different accuracy ranges. Figures 22,23, and 24 show all strongly dissimilar choices of constants found on any given task, across 53 Super Natural-Instructions tasks, and on both accuracy metrics considered throughout the work. As can be appreciated, choices of constants do not consistently predict performance in isolation.

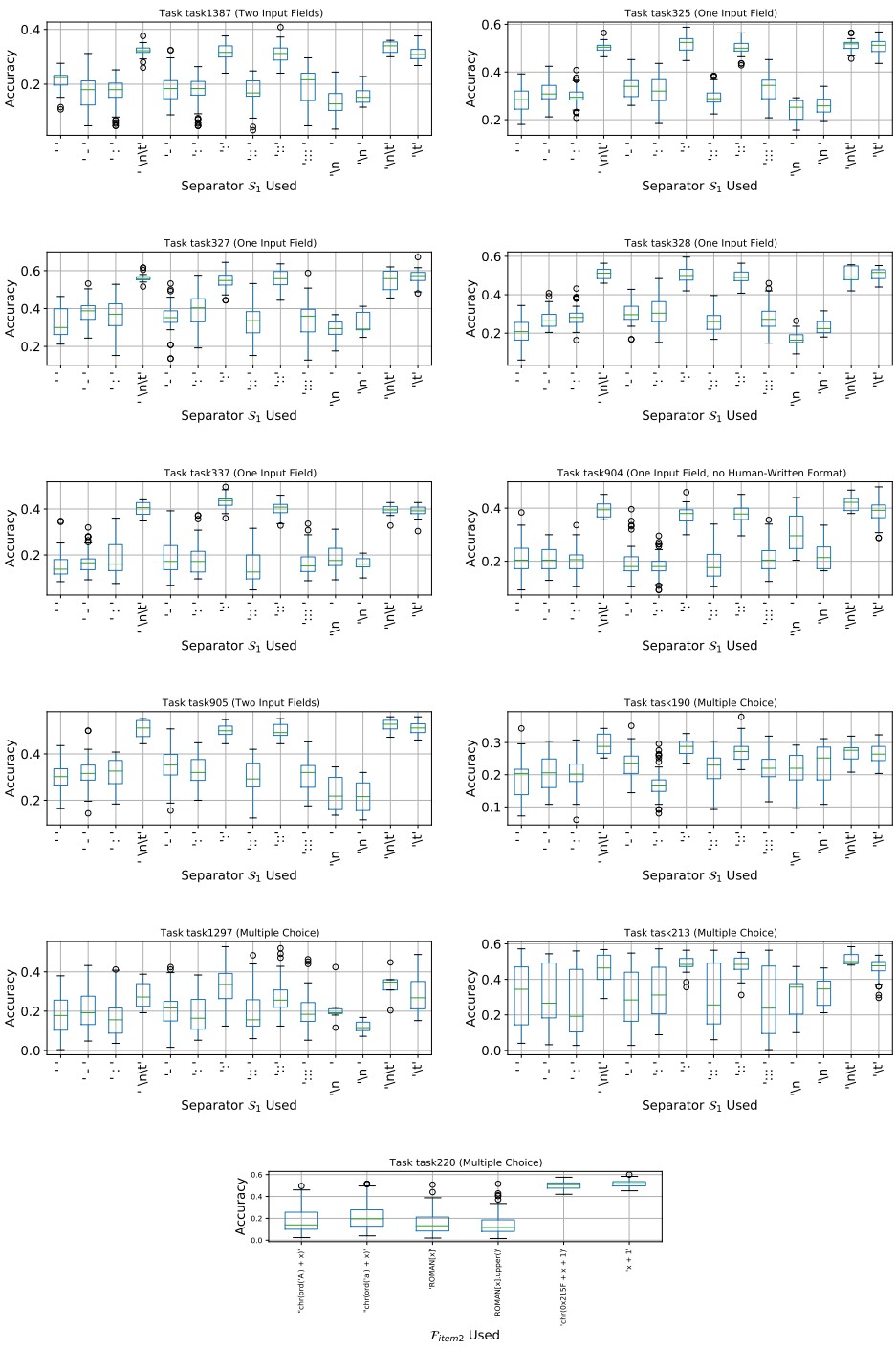

Figure 22: Variance by feature value for Llama-7B 1-shot. Evaluated 31 tasks with 500 formats each, and only plots with a significant difference between values are shown. Exact prefix matching accuracy metric.

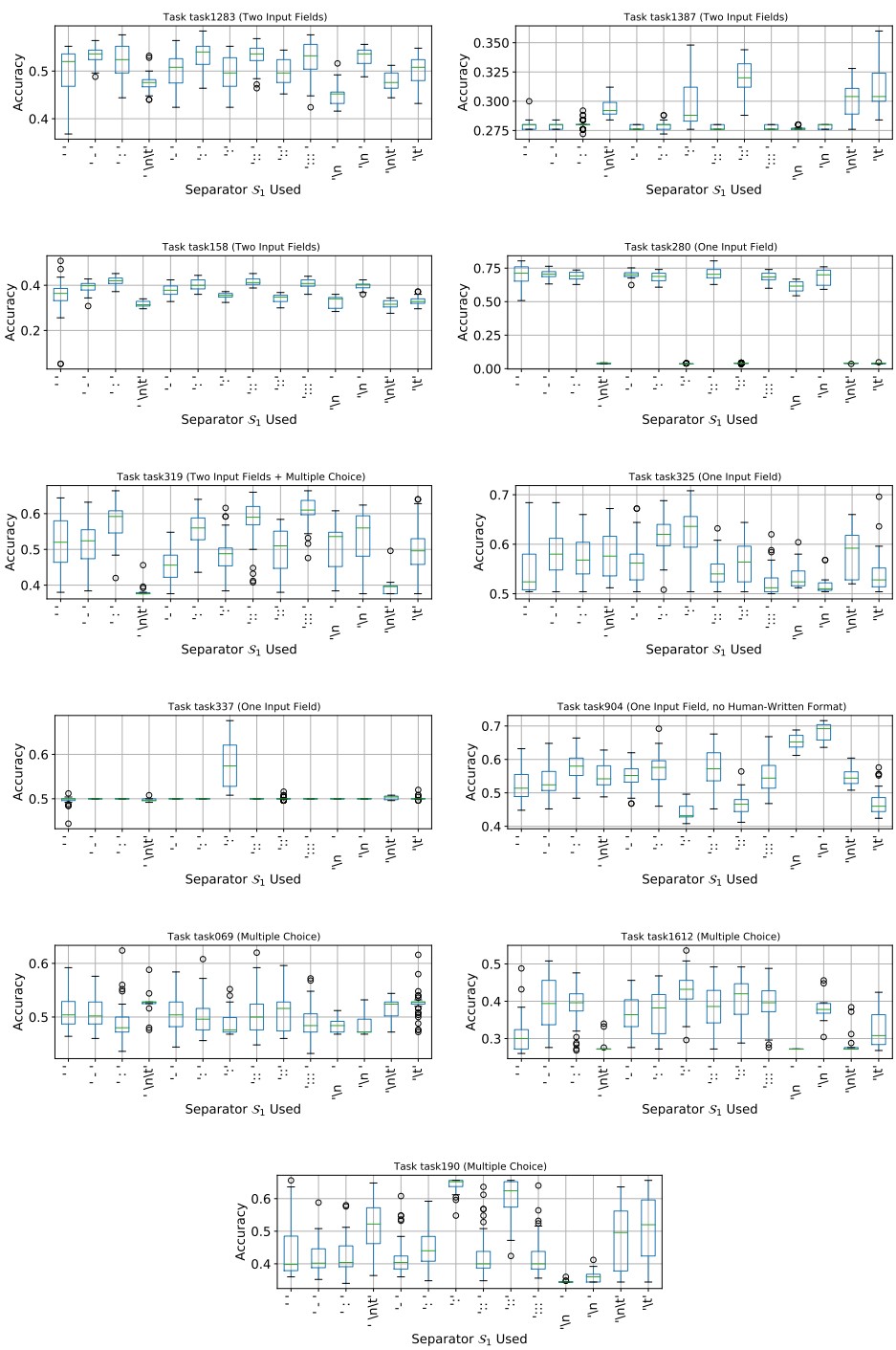

Figure 23: Variance by feature value for Llama-7B 1-shot. Evaluated 31 tasks with 500 formats each, and only plots with strongly significant differences are shown. Option ranking accuracy evaluation metric. Only $\mathcal{S}_1$ boxplots are shown here, see Figure 24 for all others.

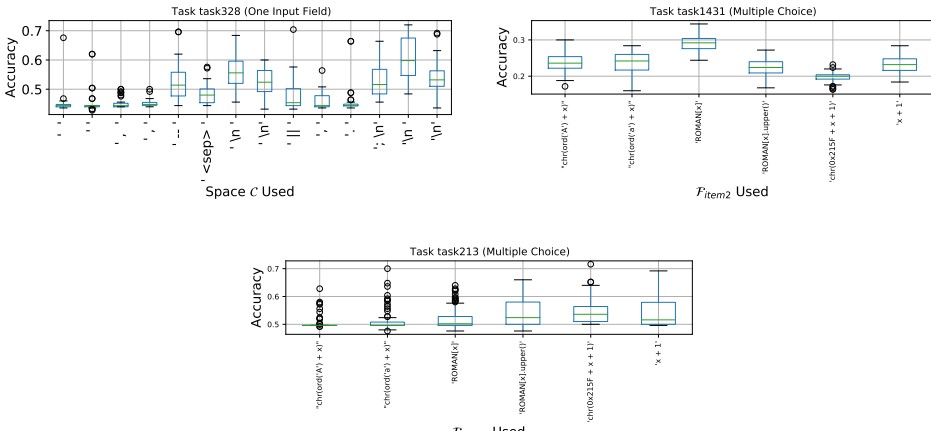

Figure 24: Variance by feature value for Llama-7B 1-shot. Evaluated 31 tasks with 500 formats each, and only plots with strongly significant differences are shown. Option ranking accuracy evaluation metric. See Figure 23 for more plots.

## B.5 THOMPSON SAMPLING RESULTS

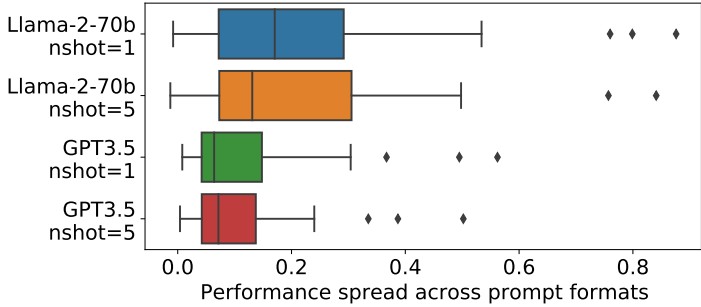

Figure 25: Spreads found using Thompson Sampling for 320 formats, budget of 40000 evaluations. LLaMA-2-70b was evaluated with option ranking, and GPT3.5 with prefix matching given that we cannot access all logits.

| Task | Model | Best Format | Worst Format | Best Acc | Worst Acc |
|------|-------|-------------|--------------|----------|-----------|
| task050 | Llama-70B | `sentence {} \|\| question {} \|\| answer {}` | `Sentence\t{}\n Question\t{}\n Answer\t{}` | 0.62 | 0.58 |
| task065[*] | Llama-70B | `Sentence<1>::  {} Sentence<3>::  {} Sentence<4>::  {} Sentence<5>::  {} \nOption A.: {} Option B.:  {} \nAnswer\n {}` | `Sentence I)\t{} \n Sentence III)\t{} \n Sentence IV)\t{} \n Sentence V)\t{} Option\t(I)::{}, Option\t(II)::{} Answer {}` | 0.88 | 0.50 |
| task069 | Llama-70B | `BEGINNING:::  {} MIDDLE A): {} , MIDDLE B): {} ENDING::  {} ANSWER:::  {}` | `BEGINNING\n {}\n MIDDLE :: {} MIDDLE <II>::  {}\n ENDING\n {}\n ANSWER\n {}` | 0.85 | 0.58 |
| task070 | Llama-70B | `Beginning:::  {}; \nMiddle\t1)\t{}; \nMiddle\t2)\t{}; \nEnding::: {}; \nAnswer:::  {}` | `Beginning {} \|\| Middle 1.:  {} \|\| Middle 2.:  {} \|\| Ending {} \|\| Answer {}` | 0.80 | 0.27 |
| task114 | Llama-70B | `sentence:  {} <sep>answer:  {}` | `Sentence:  {}\nAnswer:  {}` | 0.54 | 0.51 |
| task1186 | Llama-70B | `SYSTEM REFERENCE : {}.  ORIGINAL REFERENCE : {}.  ANSWER : {}` | `System Reference:  {}\nOriginal Reference:  {}\nAnswer:  {}` | 0.56 | 0.51 |

| task1283 | Llama-70B | system reference: {} \noriginal reference: {} \nanswer: {} | SYSTEM REFERENCE\t{}\n ORIGINAL REFERENCE\t{}\n ANSWER\t{} | 0.62 | 0.50 |
|---|---|---|---|---|---|
| task1284 | Llama-70B | SYSTEM REFERENCE\t{} \|\| ORIGINAL REFERENCE\t{} \|\| ANSWER\t{} | System Reference {} , Original Reference {} , Answer {} | 0.74 | 0.43 |
| task1297 | Llama-70B | Fact\tI.\t{} , Fact\tII.\t{} <sep>Question: {}; \n1 ) - {}; \n2 ) - {}; \n3 ) - {}; \n4 ) - {}; \n5 ) - {}; \n6 ) - {}; \n7 ) - {}; \n8 ) - {} <sep>Answer: {} | fact a )- {}\nfact b )- {} \nquestion: {} \|\| i ) : {}ii) : {}iii) : {}iv) : {}v) : {}vi) : {}vii) : {}viii) : {} \nanswer: {} | 0.88 | 0.40 |
| task133 | Llama-70B | Sentence:{} \nReason:{} \nQuestion:{} \nAnswer:{} | Sentence:: {}\n Reason:: {}\n Question:: {}\n Answer:: {} | 0.71 | 0.59 |
| task1347 | Llama-70B | SENTENCE A): {} , SENTENCE B): {} ANSWER::: {} | Sentence i){} Sentence ii){}; \nAnswer\n {} | 0.42 | 0.31 |
| task1387 | Llama-70B | Premise::{} \nHypothesis::{} \nAnswer::{} | premise {}\n hypothesis {}\n answer {} | 0.52 | 0.41 |
| task1419 | Llama-70B | problem- {} \noptions- \n <1>:: {}, <2>:: {}, <3>:: {}, <4>:: {}, <5>:: {} \nanswer- {} | Problem: {} \|\| Options: <A>{} <sep>{} <sep><C>{} <sep><D>{} <sep><E>{} \|\| Answer: {} | 0.24 | 0.23 |
| task1420 | Llama-70B | Problem\t{}\nOptions\t i):{} -- ii):{} -- iii):{} -- iv):{} -- v):{}\nAnswer\t{} | PROBLEM- {}\nOPTIONS- \n[a]{}\n [b]{}\n [c]{}\n [d]{}\n [e]{}\nANSWER- {} | 0.26 | 0.22 |
| task1421 | Llama-70B | PROBLEM:{}, OPTIONS: \na ){} -- b ){} -- c ){} -- d ){} -- e ){}, ANSWER:{} | Problem::: {} Options::: \nI )::{} \nII )::{} \nIII )::{} \nIV )::{} \nV )::{} Answer::: {} | 0.28 | 0.21 |
| task1423 | Llama-70B | problem: {} -- options: 1):: {}. 2):: {}. 3):: {}. 4):: {}. 5):: {} -- answer: {} | PROBLEM:: {}. OPTIONS:: I.- {}\n II.- {}\n III.- {}\n IV.- {}\n V.- {}. ANSWER:: {} | 0.25 | 0.20 |
| task1502 | Llama-70B | Input:{} Output:{} | input {} output {} | 0.59 | 0.45 |
| task155 | Llama-70B | SENTENCE - {}, ANSWER - {} | Sentence:: {} \nAnswer:: {} | 0.36 | 0.29 |
| task158 | Llama-70B | Sentence: {} \|\| Answer: {} | sentence::{} -- answer::{} | 0.55 | 0.49 |
| task161 | Llama-70B | Sentence\t{}\n Answer\t{} | Sentence\n {} \nAnswer\n {} | 0.45 | 0.43 |
| task1612[*] | Llama-70B | sentenceI ) - {}sentenceII ) - {} \n answer::{} | Sentence (a):{}. Sentence (b):{}\nAnswer {} | 0.64 | 0.35 |
| task162 | Llama-70B | Sentence\n {}; \nAnswer\n {} | SENTENCE \n\t{}; \nANSWER \n\t{} | 0.45 | 0.40 |
| task163 | Llama-70B | SENTENCE:{}, ANSWER:{} | SENTENCE\n {}\nANSWER\n {} | 0.47 | 0.35 |
| task1678 | Llama-70B | PROBLEM: {} , OPTIONS: \n[a] {}. [b] {}. [c] {}. [d] {}. [e] {} , ANSWER: {} | PROBLEM \n\t{} \n OPTIONS \n\t [1] {}[2] {}[3] {}[4] {}[5] {} \n ANSWER \n\t{} | 0.21 | 0.21 |
| task1724 | Llama-70B | input: {} -- output: {} | INPUT::{} , OUTPUT::{} | 0.48 | 0.48 |
| task190 | Llama-70B | Sentence a ) - {} -- Sentence b ) - {}\n Answer\n\t{} | Sentence 1.{}\nSentence 2.{}\n Answer {} | 0.66 | 0.35 |
| task213[*] | Llama-70B | TITLE::: {} \n Sentence I)::: {} , Sentence II)::: {} , Sentence III)::: {} , Sentence IV)::: {} \n CHOICES::: \ni){} \n ii){} \n ANSWER::: {} | TITLE\t{}, sentence (i)\t{} -- sentence (ii)\t{} -- sentence (iii)\t{} -- sentence (iv)\t{}, CHOICES\t \n (I):: {} (II):: {}, ANSWER\t{} | 0.99 | 0.50 |
| task214 | Llama-70B | Title: {}\n Sentence I.: {} Sentence II.: {} Sentence III.: {} Sentence IV.: {}\n Choices: \ni.::: {} \nii.::: {}\n Answer: {} | title {} \|\| sentence [1] {} sentence [2] {} sentence [3] {} sentence [4] {} \|\| choices \n I.{}; \nII.{} \|\| answer {} | 0.92 | 0.04 |
| task220 | Llama-70B | SentenceI): {}. SentenceII): {}. SentenceIII): {}. SentenceIV): {}. SentenceV): {} , Choices: \n<a>- {}\n - {} , Answer: {} | Sentence <1>: {} -- Sentence <2>: {} -- Sentence <3>: {} -- Sentence <4>: {} -- Sentence <5>: {} , Choices: \n I ){} -- II ){} , Answer: {} | 0.99 | 0.83 |
| task279 | Llama-70B | Passage:: {} \n Answer:: {} | Passage- {} Answer- {} | 0.64 | 0.49 |
| task280 | Llama-70B | Passage:: {} , Answer:: {} | PASSAGE\t{}\n ANSWER\t{} | 0.84 | 0.04 |
| task286 | Llama-70B | INPUT- {} , OUTPUT- {} | Input\t{} Output\t{} | 0.69 | 0.51 |
| task296[*] | Llama-70B | SentenceI ):: {}, SentenceII ):: {}, SentenceIII ):: {}, SentenceIV ):: {}, SentenceV ):: {}, SentenceVI ):: {}, SentenceVII ):: {}, SentenceVIII ):: {}, SentenceIX ):: {}, SentenceX ):: {}, A)\t{} , B)\t{}, Answer : {} | Sentence I.: {}; \nSentence II.: {}; \nSentence III.: {}; \nSentence IV.: {}; \nSentence V.: {}; \nSentence VI.: {}; \nSentence VII.: {}; \nSentence VIII.: {}; \nSentence IX.: {}; \nSentence X.: {} \n ::{} \|\| <ii>::{} \n Answer\n {} | 0.98 | 0.53 |

| | | | | | |
|---|---|---|---|---|---|
| task297 | Llama-70B | Sentence [a]::  {}; \nSentence [b]::  {}; \nSentence [c]::  {}; \nSentence [d]::  {}; \nSentence [e]::  {}; \nSentence [f]::  {}; \nSentence [g]::  {}; \nSentence [h]::  {}; \nSentence [i]::  {}; \nSentence [j]::  {}\n (a)::  {} (b)::  {}\n Answer {} | Sentence1:  {} Sentence2:  {} Sentence3:  {} Sentence4:  {} Sentence5:  {} Sentence6:  {} Sentence7:  {} Sentence8:  {} Sentence9:  {} Sentence10:  {} \n (A) {} (B) {} \n Answer:  {} | 0.51 | 0.03 |
| task316 | Llama-70B | passage - {} \nanswer - {} | PASSAGE::{}\nANSWER::{} | 0.51 | 0.49 |
| task317 | Llama-70B | Passage\n {}\n Answer\n {} | PASSAGE\t{}\n ANSWER\t{} | 0.83 | 0.07 |
| task319 | Llama-70B | target\n\t{}; \n{}; \nanswer\n\t{} | Target: {} {} Answer: {} | 0.77 | 0.58 |
| task320 | Llama-70B | Target: {}; \n{}; \nAnswer: {} | Target\n {} \n {} \n Answer\n {} | 0.77 | 0.58 |
| task322 | Llama-70B | COMMENT:{}\n ANSWER:{} | Comment {} -- Answer {} | 0.77 | 0.48 |
| task323 | Llama-70B | comment \n\t{}\nanswer \n\t{} | Comment {}.  Answer {} | 0.84 | 0.58 |
| task325 | Llama-70B | COMMENT\t{}; \nANSWER\t{} | Comment: {}, Answer: {} | 0.84 | 0.48 |
| task326 | Llama-70B | Comment::  {} Answer::  {} | Comment: {}, Answer: {} | 0.58 | 0.51 |
| task327 | Llama-70B | Comment {} <sep>Answer {} | Comment : {} Answer : {} | 0.88 | 0.69 |
| task328 | Llama-70B | comment:{} -- answer:{} | Comment: {} Answer: {} | 0.81 | 0.52 |
| task335 | Llama-70B | post::  {} , answer::  {} | Post: {}\nAnswer: {} | 0.52 | 0.50 |
| task337 | Llama-70B | post {} \nanswer {} | post {} answer {} | 0.85 | 0.63 |
| task385 | Llama-70B | CONTEXT {} -- QUESTION {} -- OPTIONS \n1.- {}, 2.- {}, 3.- {} -- ANSWER {} | context \n\t{} \nquestion \n\t{} \noptions \n\t 1 )::{}. 2 )::{}.  3 )::{} \nanswer \n\t{} | 0.38 | 0.11 |
| task580 | Llama-70B | Context- {}\n Question- {}\n Options- \na) {} -- b) {} -- c) {}\n Answer- {} | Context {}, Question {}, Options \n I ) - {} \nII ) - {} \nIII ) - {}, Answer {} | 0.78 | 0.40 |
| task607 | Llama-70B | INPUT \n\t{} \n OUTPUT \n\t{} | Input : {}.  Output : {} | 0.68 | 0.51 |
| task608 | Llama-70B | input: {}.  output: {} | INPUT \n\t{}\nOUTPUT \n\t{} | 0.71 | 0.48 |
| task609 | Llama-70B | Input \n\t{} \n Output \n\t{} | Input : {}, Output : {} | 0.71 | 0.51 |
| task904 | Llama-70B | input:::  {} , output:::  {} | INPUT::{}\n OUTPUT::{} | 0.71 | 0.55 |
| task905 | Llama-70B | Tweet::{} || Label::{} || Answer::{} | TWEET: {} \n LABEL: {} \n ANSWER: {} | 0.68 | 0.50 |
| task050 | GPT3.5 | Sentence\n\t{} \n Question\n\t{} \n Answer\n\t{} | sentence: {} , question: {} , answer: {} | 0.67 | 0.61 |
| task065 | GPT3.5 | SENTENCEi.:::  {} \nSENTENCEiii.:::  {} \nSENTENCEiv.:::  {} \nSENTENCEv.:::  {} \nOPTION [1]\t{}OPTION [2]\t{} \nANSWER\t{} | Sentence 1 ) : {} , Sentence 3 ) : {} , Sentence 4 ) : {} , Sentence 5 ) : {} <sep>Option (i): {} <sep>Option (ii): {} <sep>Answer- {} | 0.82 | 0.33 |
| task069 | GPT3.5 | BEGINNING: {}\n MIDDLE\t(I):: {} \n MIDDLE\t(II)::: {}\n ENDING: {}\n ANSWER: {} | Beginning {} || Middle [a]- {} \n Middle [b]- {} || Ending {} || Answer {} | 0.83 | 0.65 |
| task070[*] | GPT3.5 | Beginning : {}\nMiddle (I): {} -- Middle (II): {}\nEnding : {}\nAnswer : {} | Beginning\t{} <sep>Middle i){} || Middle ii){} <sep>Ending\t{} <sep>Answer\t{} | 0.70 | 0.49 |
| task114 | GPT3.5 | sentence \n\t{}\nanswer \n\t{} | SENTENCE: {}.  ANSWER: {} | 0.67 | 0.63 |
| task1186 | GPT3.5 | system reference : {} , original reference : {} , answer : {} | System Reference: {}\nOriginal Reference: {}\nAnswer: {} | 0.53 | 0.51 |
| task1283 | GPT3.5 | SYSTEM REFERENCE : {} , ORIGINAL REFERENCE : {} , ANSWER : {} | System Reference\n {} \nOriginal Reference\n {} \nAnswer\n {} | 0.57 | 0.50 |
| task1284 | GPT3.5 | System Reference \n\t{}\n Original Reference \n\t{}\n Answer \n\t{} | system reference::: {} -- original reference::: {} -- answer::: {} | 0.63 | 0.57 |
| task1297[*] | GPT3.5 | Fact<1>: {} -- Fact<2>: {}; \nQuestion::{}\nI):: {} II):: {} III):: {} IV):: {} V):: {} VI):: {} VII):: {} VIII):: {}; \nAnswer::{} | fact\ta. : {} || fact\tb. : {}\n question : {} \na. - {}b. - {}c. - {}d. - {}e. - {}f. - {}g. - {}h. - {}\n answer : {} | 0.84 | 0.72 |
| task133 | GPT3.5 | Sentence::{} -- Reason::{} -- Question::{} -- Answer::{} | Sentence - {}; \nReason - {}; \nQuestion - {}; \nAnswer - {} | 0.69 | 0.64 |
| task1347[*] | GPT3.5 | SentenceI.::  {}\n SentenceII.::  {} Answer\t{} | Sentence (I){} Sentence (II){} \nAnswer::{} | 0.46 | 0.42 |
| task1387 | GPT3.5 | Premise:{} , Hypothesis:{} , Answer:{} | PREMISE::  {}; \nHYPOTHESIS:: {}; \nANSWER:: {} | 0.47 | 0.44 |
| task1419 | GPT3.5 | Problem - {} || Options - \n[A]\t{}.  [B]\t{}.  [C]\t{}. [D]\t{}.  [E]\t{} || Answer - {} | PROBLEM- {}\nOPTIONS- 1 ) - {} -- 2 ) - {} -- 3 ) - {} -- 4 ) - {} -- 5 ) - {}\nANSWER- {} | 0.24 | 0.20 |

| task1420[*] | GPT3.5 | problem- {} \n options- (I) - {}(II) - {}(III) - {}(IV) - {}(V) - {} \n answer- {} | Problem:: {} -- Options:: \n[i]: {}; \n[ii]: {}; \n[iii]: {}; \n[iv]: {}; \n[v]: {} -- Answer:: {} | 0.24 | 0.09 |
|---|---|---|---|---|---|
| task1421 | GPT3.5 | Problem \n\t{}\n Options \n\t \nA)::{} -- B)::{} -- C)::{} -- D)::{} -- E)::{}\n Answer \n\t{} | PROBLEM::{}, OPTIONS:: \n[a]: {} \n[b]: {} \n[c]: {} \n[d]: {} \n[e]: {}, ANSWER::{} | 0.30 | 0.01 |
| task1423 | GPT3.5 | PROBLEM: {}; \nOPTIONS: \n a ) {} \n b ) {} \n c ) {} \n d ) {} \n e ) {}; \nANSWER: {} | Problem:: {} -- Options:: \n[i]: {}; \n[ii]: {}; \n[iii]: {}; \n[iv]: {}; \n[v]: {} -- Answer:: {} | 0.30 | 0.10 |
| task1502 | GPT3.5 | input\n {}\noutput\n {} | input::{} -- output::{} | 0.54 | 0.48 |
| task155 | GPT3.5 | SENTENCE::{} -- ANSWER::{} | SENTENCE {} ANSWER {} | 0.48 | 0.40 |
| task158 | GPT3.5 | Sentence:{} Answer:{} | SENTENCE\t{} , ANSWER\t{} | 0.63 | 0.60 |
| task161 | GPT3.5 | sentence\t{} answer\t{} | Sentence- {}\nAnswer- {} | 0.40 | 0.31 |
| task1612 | GPT3.5 | sentence 1 ): {} || sentence 2 ): {}, answer::{} | Sentence I ) : {} \nSentence II ) : {} , Answer {} | 0.66 | 0.54 |
| task162 | GPT3.5 | Sentence- {} -- Answer- {} | SENTENCE\t{}; \nANSWER\t{} | 0.37 | 0.31 |
| task163 | GPT3.5 | Sentence- {}; \nAnswer- {} | sentence: {}\nanswer: {} | 0.43 | 0.39 |
| task1678[*] | GPT3.5 | Problem - {} Options - \n (I) {} <sep>(II) {} <sep>(III) {} <sep>(IV) {} <sep>(V) {} Answer - {} | PROBLEM::{} \n OPTIONS:: \n[I] {} , [II] {} , [III] {} , [IV] {} , [V] {} \n ANSWER::{} | 0.24 | 0.09 |
| task1724 | GPT3.5 | INPUT::{} || OUTPUT::{} | INPUT: {} OUTPUT: {} | 0.63 | 0.57 |
| task190 | GPT3.5 | Sentence (I) - {}; \nSentence (II) - {}; \nAnswer {} | Sentence<1>: {}\nSentence<2>: {}, Answer : {} | 0.29 | 0.16 |
| task213 | GPT3.5 | Title: {} <sep>Sentence <1>: {} || Sentence <2>: {} || Sentence <3>: {} || Sentence <4>: {} <sep>Choices: \n- {} <sep><ii>- {} <sep>Answer: {} | TITLE:{}, sentence\t[a]:{} -- sentence\t[b]:{} -- sentence\t[c]:{} -- sentence\t[d]:{}, CHOICES: \n[i]:{} [ii]:{}, ANSWER:{} | 0.98 | 0.81 |
| task214[*] | GPT3.5 | title: {} \nSentencei): {}, Sentenceii): {}, Sentenceiii): {}, Sentenceiv): {} \nchoices: \n- {} , <II>- {} \nanswer: {} | Title::{} , Sentence\ta )::{} , Sentence\tb )::{} , Sentence\tc )::{} , Sentence\td )::{} , Choices:: [i]. {}[ii]. {} , Answer::{} | 0.97 | 0.41 |
| task220 | GPT3.5 | Sentence [1]: {} \n Sentence [2]: {} \n Sentence [3]: {} \n Sentence [4]: {} \n Sentence [5]: {} -- Choices: \n <A>::{}\n ::{} -- Answer: {} | Sentence\ti.: {} || Sentence\tii.: {} || Sentence\tiii.: {} || Sentence\tiv.: {} || Sentence\tv.: {} Choices: \n[i]:{} || [ii]:{} Answer: {} | 0.98 | 0.79 |
| task279 | GPT3.5 | passage {} , answer {} | Passage:{}, Answer:{} | 0.58 | 0.56 |
| task280 | GPT3.5 | Passage - {} <sep>Answer - {} | Passage {} -- Answer {} | 0.85 | 0.80 |
| task286 | GPT3.5 | Input- {}\n Output- {} | INPUT:: {} OUTPUT:: {} | 0.72 | 0.69 |
| task296 | GPT3.5 | Sentence a ) - {} \nSentence b ) - {} \nSentence c ) - {} \nSentence d ) - {} \nSentence e ) - {} \nSentence f ) - {} \nSentence g ) - {} \nSentence h ) - {} \nSentence i ) - {} \nSentence j ) - {} \n<A>- {}, - {} \nAnswer::: {} | Sentence\t(I)- {} -- Sentence\t(II)- {} -- Sentence\t(III)- {} -- Sentence\t(IV)- {} -- Sentence\t(V)- {} -- Sentence\t(VI)- {} -- Sentence\t(VII)- {} -- Sentence\t(VIII)- {} -- Sentence\t(IX)- {} -- Sentence\t(X)- {}, I )::{}, II )::{}, Answer:{} | 0.95 | 0.68 |
| task297 | GPT3.5 | Sentence (A): {}; \nSentence (B): {}; \nSentence (C): {}; \nSentence (D): {}; \nSentence (E): {}; \nSentence (F): {}; \nSentence (G): {}; \nSentence (H): {}; \nSentence (I): {}; \nSentence (J): {}; \n::{} <sep><II>::{}; \nAnswer {} | SENTENCE\t(A) : {}; \nSENTENCE\t(B) : {}; \nSENTENCE\t(C) : {}; \nSENTENCE\t(D) : {}; \nSENTENCE\t(E) : {}; \nSENTENCE\t(F) : {}; \nSENTENCE\t(G) : {}; \nSENTENCE\t(H) : {}; \nSENTENCE\t(I) : {}; \nSENTENCE\t(J) : {} -- I) : {} II) : {} -- ANSWER: {} | 0.36 | 0.06 |
| task316 | GPT3.5 | passage\n\t{} \nanswer\n\t{} | Passage- {} Answer- {} | 0.49 | 0.48 |
| task317 | GPT3.5 | passage: {} answer: {} | passage {} answer {} | 0.75 | 0.70 |
| task319 | GPT3.5 | TARGET:: {} \n{} \nANSWER:: {} | Target: {}\n{}\nAnswer: {} | 0.66 | 0.62 |
| task320 | GPT3.5 | Target::: {} \n{} \nAnswer::: {} | target {} -- {} -- answer {} | 0.73 | 0.68 |
| task322 | GPT3.5 | Comment: {} -- Answer: {} | comment::{} answer::{} | 0.84 | 0.83 |

| | | | | | |
|---|---|---|---|---|---|
| task323 | GPT3.5 | `COMMENT {} <sep>ANSWER {}` | `comment - {}\nanswer - {}` | 0.73 | 0.63 |
| task325 | GPT3.5 | `comment {}; \nanswer {}` | `Comment\n {}\n Answer\n {}` | 0.81 | 0.74 |
| task326 | GPT3.5 | `comment : {} , answer : {}` | `Comment- {} Answer- {}` | 0.67 | 0.66 |
| task327 | GPT3.5 | `comment:{} \nanswer:{}` | `Comment - {} \|\| Answer - {}` | 0.86 | 0.82 |
| task328 | GPT3.5 | `Comment - {}. Answer - {}` | `COMMENT {} <sep>ANSWER {}` | 0.77 | 0.72 |
| task335 | GPT3.5 | `Post\n\t{} \nAnswer\n\t{}` | `post - {} -- answer - {}` | 0.37 | 0.20 |
| task337 | GPT3.5 | `Post::{}; \nAnswer::{}` | `Post {}, Answer {}` | 0.57 | 0.51 |
| task385 | GPT3.5 | `Context:: {}; \nQuestion:: {}; \nOptions:: a)::: {}\n b)::: {}\n c)::: {}; \nAnswer:: {}` | `Context:{}. Question:{}. Options:\n[i] {}[ii] {}[iii] {}. Answer:{}` | 0.46 | 0.10 |
| task580 | GPT3.5 | `Context:: {} \|\| Question:: {} \|\| Options:: \n A)- {} B)- {} C)- {} \|\| Answer:: {}` | `CONTEXT:: {} -- QUESTION:: {} -- OPTIONS:: \n[I] - {}. [II] - {}. [III] - {} -- ANSWER:: {}` | 0.74 | 0.65 |
| task607 | GPT3.5 | `input {} \|\| output {}` | `INPUT:: {}\nOUTPUT:: {}` | 0.70 | 0.68 |
| task608 | GPT3.5 | `Input:: {} Output:: {}` | `Input\n {} \nOutput\n {}` | 0.60 | 0.52 |
| task609 | GPT3.5 | `INPUT\n {} \nOUTPUT\n {}` | `Input {}\nOutput {}` | 0.70 | 0.68 |
| task904 | GPT3.5 | `INPUT : {}\nOUTPUT : {}` | `Input:: {}, Output:: {}` | 0.66 | 0.61 |
| task905 | GPT3.5 | `Tweet:{} , Label:{} , Answer:{}` | `TWEET {} \|\| LABEL {} \|\| ANSWER {}` | 0.72 | 0.63 |

Table 6: Best and worst formats found by using Thompson Sampling on 320 formats on Llama-70B and GPT3.5 , $E = 40000$, $B = 10$. Tasks marked with an asterisk[*] indicate a format where the Roman numerals used correspond to its Unicode characters (starting at 0x215F), visualized as I, II, III, IV, . . ..

## C    LIMITATIONS

As defined by our grammar, all equivalent formats are semantically equivalent to human read-ers. However, some of them are more likely to be used by humans than others. Spaces and separators are inspired from naturally-occurring formats, but some values are more un-usual, such as the spacing `<sep>` or the separator `::`. Contextual restrictions enable disal-lowing undesired combinations of e.g. spaces and separators. However, formats may have multiple valid parses, and some may be more prone than others to unnatural character combi-nations. For example, let a data sample be `Passage:   Lorem ipsum dolor sit amet.   Answer:   Yes`. Depending on if we consider the full stop `.` to be part of the passage or the format, we may parse it as $B_2^{(2)}(B_1(\texttt{Passage},': \ ',id),B_1(\texttt{Answer},': \ ',id),' \ ')$ or $B_2^{(2)}(B_1(\texttt{Passage},': \ ',id),B_1(\texttt{Answer},': \ ',id),'. \ ')$. In this work, we choose the for-mer parsing throughout tasks to ensure full sentences. This sometimes[7] leads equivalent for-mats to have a less usual, yet trivially semantically equivalent resulting character combinations, e.g. $B_2^{(2)}(B_1(\texttt{Passage},': \ ',id),B_1(\texttt{Answer},': \ ',id),'; \ ')$. This last format would have the following string form on the example above: `Passage:   Lorem ipsum dolor sit amet.; Answer:   Yes`. We observe high performance spread both in these cases and beyond them. Contextual relations may also restrict these cases if desired by the end user.

Additionally, we focus our evaluation on tasks that have reasonably short input instructions and input field length (see task selection details in B.1). Future work may investigate on how input length affects final performance.

---

[7]Less than 20% of cases, based on a manual inspection of 10 formats across 20 tasks.

