# OpenReview forum: "Quantifying Language Models' Sensitivity to Spurious Features in Prompt Design or: How I learned to start worrying about prompt formatting"
_ICLR.cc/2024/Conference — ICLR 2024 poster_

### Official Review · Reviewer_c9z9 · 2023-10-31

**Soundness:** 3 good
**Presentation:** 3 good
**Contribution:** 3 good
**Rating:** 6
**Confidence:** 2

**Summary:**

This paper explores the sensitivity of large language models (LLMs) to prompt formatting choices and investigates the impact of prompt design on LLM performance in few-shot (especially few-shot classification) settings. The authors find that even subtle changes in prompt formatting can have a significant effect on model behavior, with performance differences of up to 76 accuracy points. This sensitivity remains even when the model size or the number of few-shot examples is increased.
The authors argue that evaluating LLMs with a single prompt format is inadequate and propose reporting a range of performance across plausible formats.
They also demonstrate that format performance weakly correlates between models, questioning the validity of comparing models using a fixed prompt format. To facilitate systematic analysis, the authors introduce an algorithm called FORMATSPREAD, which quickly evaluates a sampled set of prompt formats for a given task. The paper emphasizes the importance of reporting and considering prompt format variations when comparing models and highlights the impact of formatting choices on model behavior by extensive experiments.

**Strengths:**

1. The paper investigates a critical issue for Large Language Models (LLMs), specifically the impact of formatting on the few-shot examples used in prompts.

2. The assertion that "The performance of Large Language Models (LLMs) is highly sensitive to prompt formatting choices, especially in few-shot settings," is substantiated by numerous experiments on few-shot classification tasks (e.g., Super-Natural Instructions) and short text generation tasks, such as identifying the second letter of a word, performing arithmetic, or responding with a synonym for a given word.

3. In Section 3, the authors formally define the "grammar" of plausible prompt formats, thereby making the problem formulation more rigorous.

**Weaknesses:**

1. The paper primarily substantiates its core claim that "Performance of large language models (LLMs) is highly sensitive to prompt formatting choices, particularly in few-shot settings," through experiments in classification tasks. However, the scope of the experiments does not extend to the frequently utilized capability of LLMs for long text generation. While short text generation tasks (such as identifying the second letter of a word, adding numbers, or responding with a synonym) are discussed in the appendix, these do not fully capture the important aspect of long text generation. Hence, I suggest that the authors either explicitly state that these findings are specifically under the context of classification tasks or conduct additional experiments on long-text generation to avoid any potential overclaim or misleading interpretation.

2. For Figure 2, I recommend that the authors include the Pearson correlation coefficient directly on the figure for a more comprehensive representation of the data.

Overall, I believe this paper studies an important question. If the authors can address my concerns, I would consider increasing my score.

**Questions:**

1. In Section 3.2, the performance spread is quantified as the difference between the maximum and minimum values. I suggest that a more comprehensive approach would be to report both the range (max - min) and the standard deviation, providing a fuller understanding of the data distribution.

2. In the experimental section, the authors explain that classification tasks were selected for their ease of automatic evaluation. I am curious about the challenges associated with measuring performance for text generation tasks, especially considering the benchmarks that have been proposed recently.

---

> ### Author Response · Authors · 2023-11-20
> **Response to Reviewer c9z9**
>
> We thank the reviewer for their thoughtful review! We are pleased that they pointed out our paper studies “a critical issue for LLMs”, with a rigorous problem formulation and substantiated with numerous experiments on few-shot classification tasks (e.g., Super-Natural Instructions) and short text generation tasks.
>
> We address all comments below, and **we believe that the two new, additional experiments** *(an experiment on performance variance on 17 new sentence-level open text generation tasks; and an analysis of 30 classification tasks quantifying the dispersion of the accuracy distribution using standard deviation on 500 formats evaluated)* **resolve all the reviewers’ concerns**!
>
> - ***“A more comprehensive approach would be to report both the range plus the standard deviation”*** We agree that including standard deviation gives a more comprehensive understanding of the distribution, and we have added analysis to Appendix B.2 (see “Characterizing a model’s accuracy distribution beyond spread” and Figure 20) showing the distribution of accuracy across 500 formats for 30 different tasks. Plots also include the standard deviation, which varies from stdev=0.31 to 0.02 across the tasks considered (median stdev=0.04). We would like to emphasize that reporting the spread (i.e. the distribution range) is already a much more comprehensive report than the single-format estimation that is currently the norm, and that efficiently estimating the standard deviation of a distribution is a priori a much more computationally expensive problem. This would require precisely computing the accuracy of possibly hundreds of formats, and would render our current bandit formulation unsuitable. We nonetheless believe that efficiently computing the standard deviation of such a distribution of accuracies is an exciting question for future work!
>
> - ***“Why classification tasks? What are the challenges associated with measuring performance in long generation tasks?”*** We focused primarily on classification tasks since they unambiguously showcase clearly that performance spread can be a serious concern, given that they are generally much more straightforward to evaluate than text generation metrics. Text generation evaluation is notoriously more difficult, noisy, and subjective than simple classification metrics. Text generation metrics remain an active area of research, and metrics are often task-specific (e.g. a summarization metric) which would hinder us from using the same metric across tasks. Moreover, evaluating text generation often requires a multifaceted approach (e.g. fluency, helpfulness, similarity with the recorded human response) and we specifically wanted to avoid conflating the establishment of the spread discussion with the open problem of text generation evaluation. We nonetheless conducted additional experiments on sentence-length text generation problems; see next question for details!
>
> - ***“I suggest that the authors either explicitly state that these findings are specifically under the context of classification tasks or conduct additional experiments on long-text generation”*** We agree with the reviewer that longer text generation is an important setting to evaluate for LLM usage, and thus we are included additional experiments for text generation using BertScore and ROUGE-L as metrics in App. B.2 (“Experiments with continuous metrics in open-ended text generation tasks”, also Fig. 19), showing that the issue of performance variance across formats still holds: we computed spread on 10 randomly sampled formats on 17 open text generation tasks with LLaMA-2-7B, and found that performance variance remains regardless of the metric and the number of n-shots considered, with LLaMA-2-7B 5-shot having 25% of tasks with a ROUGE-L spread ≥ 0.098, and a BERTScore spread ≥ 0.09. ROUGE-L is the metric used in the original Super-NaturalInstructions tasks, but is known to have a preference for lexical similarity; therefore, we complement results with BertScore, a model-based metric (which, like all model-based open generation metrics, is more robust to changes in vocabulary but that will express Bert’sinductive bias). We focus our new experiments on evaluating sentence-long text generation tasks, since longer text generation experiments are significantly more computationally intensive, given that they require processing a significantly longer input including the solved few-shot examples, plus generating a long text. We will carefully qualify the extent of our experiments across the manuscript: we also added a Limitations section (Appendix C) in this same vein.
>
> - ***“Pearson correlation coefficient in Figure 2?”*** We have now included the Pearson correlation coefficient in all scatter plots, including appendix.
>
> We believe this resolves the concerns the reviewer flagged and we thank them again for their thorough review!

---

> > ### Comment · Reviewer_c9z9 · 2023-11-22
> >
> > Dear Authors,
> >
> > Thank you all for your responses.  There are no other serious problems on my side. Consequently, I have adjusted my rating from 5 to 6.
> >
> > Best,

---

### Official Review · Reviewer_fgsZ · 2023-11-01

**Soundness:** 3 good
**Presentation:** 3 good
**Contribution:** 4 excellent
**Rating:** 8
**Confidence:** 4

**Summary:**

Authors claim that in recent trends, LLMs are the inevitable choice for language technologies; their sensitivity to prompt formatting results in critical issues. The sensitivity remains critical even after increasing model size, usage of in-context learning, and instruction-tuning. Given this situation, authors suggest evaluating the performance of LLMs with various prompt formatting, not with a single prompt format. The authors also report that format performance across LLMs has a low correlation, which supports the multi-prompt format evaluation of LLMs. To facilitate the multi-prompt format, the authors propose an analysis method, FORMATSPREAD, that evaluates a set of plausible prompt formats for a given task. The proposed method induces the concept of semantically equivalent formats and measures the performance gap among LLMs queried with the different formats but in a semantically equivalent set, procured by the help of Bayesian Optimization.

**Strengths:**

- Focus on the subject that has not taken much attention from the community but should be addressed for robust application of LLMs
- Evaluating LLMs over the prompt distribution provides a more informative understanding of the model's performance and robustness than evaluating only with a single prompt.
- The proposed method can be utilized on API-gated models, with no need for access to model weights.

**Weaknesses:**

- The authors cast the problem of searching prompt space as a bandit problem. However, many current important applications of LLM assume multi-turn conversation between the user and LLM, which is highly dependent on the conversation history.
- In Algorithm 1, the formulation assumes the reward of each arm is success or failure, but in the NLP field, there are many important tasks where the output of LM cannot be determined between success or failure (for example, a task needs human alignment). Does the formulation still hold for the tasks that cannot be evaluated in discrete value?

**Questions:**

- As the search space of the prompt is intractable, a good initial point (in this case, a prompt) would be crucial for successful entire optimization (as authors assume non-adversarial user). How can we be convinced that we start from a good initial point?

---

> ### Author Response · Authors · 2023-11-20
> **Response to Reviewer fgsZ**
>
> We thank the reviewer for their insightful and encouraging feedback! We are glad they pointed out our work covers a “subject that has not taken much attention from the community but should be addressed for robust application of LLMs”, with a method that “can be utilized on API-gated models” and gives a “more informative understanding of the model's performance and robustness”. We address all questions below.
>
> - ***“Does the formulation still hold for the tasks that cannot be evaluated in discrete value?”*** It does! Thompson Sampling still works for continuous rewards (see Algorithm 1 of Chappelle & Li, 2011; Bernoulli bandits are shown in Algorithm 2). We focused our explanation on the exact setting we used throughout the paper for ease of reading, but the only requirement to extend it to continuous rewards would be to provide a more suitable reward distribution to sample from instead of sampling from a Beta (e.g. a Gaussian).
>
> - ***“A good initial point (a prompt) would be crucial for a successful entire optimization. How can we be convinced that we start from a good point?”*** We are not using a local search algorithm, where an initial point would strongly influence the trajectory of the search; instead, our approach samples random formats by assigning different values to all the constants defined in the grammar. The only influence that the initial prompt has on the search is that if such a prompt is available for a task in our evaluation data, we use it to induce the prompt format grammar for that task.
>
> - ***“current important applications of LLM assume multi-turn conversation between the user and LLM, which is highly dependent on the conversation history”*** We fully agree! Formatting brittleness issues may still stand in these settings, as one can consider each partial conversation as an individual test point to be analyzed in the context of FormatSpread. Conducting a sensitivity experiment with humans in the loop would be difficult because the user behavior might also change in response to system behavior. However, we hypothesize that this sensitivity may be even more detrimental in these multi-turn settings, as small differences in behavior might compound turn after turn. This would be an exciting future direction to test LLM robustness beyond the few-shot setting we focus on!

---

### Official Review · Reviewer_RFEF · 2023-11-08

**Soundness:** 3 good
**Presentation:** 3 good
**Contribution:** 3 good
**Rating:** 6
**Confidence:** 3

**Summary:**

This paper explores a very interesting question, which is the impact of different prompt formats of LLMs on the accuracy of downstream tasks. The authors found that this impact is significant to some extent and present a new algorithm called FORMATSPREAD, which can estimate the performance spread of different prompt formatting choices. FORMATSPREAD efficiently searches the space of plausible prompt formats within a specified computational budget.

**Strengths:**

- The issue discussed in this paper, i.e., LLM evaluation should not be limited to a single specific prompt, has good informativeness for the community.
- This paper is well written and easy to understand.
- The proposed construction of grammar rules for different prompt formats and the design of sensitivity evaluation are quite ingenious.

**Weaknesses:**

As the authors claim that LLMs are extremely sensitive to subtle changes in prompt formatting, with performance differences of up to 76 accuracy points, which is quite surprising. It is necessary to conduct a more in-depth analysis of these somewhat counterintuitive conclusions. For example,
1. Is the difference in  prompt formats the only influencing factor, or do other confounders exist, such as the content length of in-context, different tokenize methods?
2. It is difficult to predict the impact on specific task sensitivities. How can we analyze which types of tasks are more susceptible to prompt format influences, rather than just conducting sensitivity evaluations? This requires further explanation.

**Questions:**

1. Add more analysis about cofounders, such as content length in the examplars, different tokenize methods, etc.
2. Add further explanation of which specific tasks are more susceptible of changes in prompt formatting.

---

> ### Author Response · Authors · 2023-11-20
> **Response to Reviewer RFEF**
>
> We thank the reviewer for their detailed feedback! We are glad they mentioned that we explore “a very interesting question” and that our “proposed construction of grammar rules [...] and the design of sensitivity evaluation are quite ingenious”. We address all questions below.
>
>
> - ***“Is the difference in prompt formats the only influencing factor, or do other confounders exist, such as the content length of in-context, different tokenize methods?”*** This is a great suggestion. While our work focuses on measuring the sensitivity to prompt formatting in particular, our analysis in Section 4.2 could definitely be adapted to features like the length of in-context examples. We don’t claim prompt formatting to be the sole influencing factor on performance variation, but rather a compelling example of spurious features that can seriously affect final performance. Nonetheless, **this question inspired us to add an analysis on the impact of in-context length in Appendix B.2** (“On factors influencing performance variance besides prompt formatting”), where we found a negligible correlation between the spread and both the average and the standard deviation of prompt length for a task: correlation was $r=0.228$ ($p=1.4\times 10^{-7}$) for mean prompt length when using exact matching prefix as accuracy metric, and $r=-0.022$ ($p=0.615$) for option ranking metric, see App B.2 for all details. Analyzing the impact of a specific tokenizer exceeds the scope of this work, since decoupling the influence of a model’s tokenizer vs. other factors like the data it was trained on would require training several models with identical training procedures but different tokenizers.
>
> - ***“How can we analyze which types of tasks are more susceptible to prompt format influences, rather than just conducting sensitivity evaluations?”*** This is an interesting question, and one we focused on as part of our preliminary explorations. We did not find specific factors that can faithfully predict a priori the sensitivity of a model on a given task. For example, we did not find that multiple choice questions had any significant difference from regular tasks with two fields (e.g. “Input: <text> Output: <text>”) or more. However, digging more into why certain tasks result in more sensitivity is an exciting direction for future work!
>
> - ***On the counterintuitiveness of our sensitivity results.*** Sensitivity to spurious features in prompt design is understudied and often not taken into account; much existing research and practice uses LLMs with the assumption they are robust to these features, so we believe that quantifying this sensitivity is crucial to give it its deserved visibility! That being said, we do not believe that our results are necessarily counterintuitive to what we know about LLMs. For example, few-shot examples’ ordering (which doesn’t drastically change the semantics of a prompt) may affect final results (Lu et al., 2022).  As models with multiple billions of parameters, it is not unreasonable to expect their distribution to have sharp changes and discontinuities: we do however hope that FormatSpread sparks discussions on new training objectives that provide robustness guarantees to spurious prompt changes!

---

### Author Response · Authors · 2023-11-20
**General Response**

We thank the reviewers for their thoughtful feedback! We are encouraged that they found our paper “explores a very interesting question” (R1), tackling a “critical issue” (R3) “that has not taken much attention but should be addressed for robust application of LLMs” (R2). We are honored to hear that reviewers found our methodology to be “quite ingenious” (R1), with our grammar definition enabling a more rigorous problem formulation (R3), and our sensitivity evaluation method being so general that “it can be utilized on API-gated models” (R2).

Besides our method contributions, reviewers highlighted that our claims are “substantiated by numerous experiments on few-shot classification tasks and short text generation tasks” (R3).

Based on reviewers’ feedback, **we have conducted and added three new experiments/analyses** in Appendix B.2:
- We added an experiment that computes spread on 10 randomly sampled formats on 17 open text generation tasks with LLaMA-2-7B, measured with well-known metrics (ROUGE-L and BERTScore) to show that the performance variation issue may still hold beyond classification accuracy (which remains our main focus, due to its reliability as a metric when applicable). In a nutshell, performance variance remains regardless of the metric and the number of n-shots considered, with LLaMA-2-7B 5-shot having 25% of tasks with a ROUGE-L spread of 0.098 or higher, and a BERTScore spread of 0.09 or higher. **This new experiment brings our total of tasks analyzed to 53+17+10=80 tasks!** We will include the evaluation of all remaining models for the camera ready. See details in “Experiments with continuous metrics in open-ended text generation tasks.” in Appendix B.2., and Figure 19.
- We added plots of the entire distributions of accuracy, evaluating 30 tasks on 500 formats, and computing their standard deviation to give a more comprehensive view of the phenomenon beyond spread (i.e. distribution range). We found the standard deviation to often be significantly large, suggesting that our reported spread corresponds to more than just outliers in the distribution. For example, it would not be unusual to find high accuracy variance even when randomly sampling just two formats. See “Characterizing a model’s accuracy distribution beyond spread” in Appendix B.2 and Figure 20.
- We included an analysis of how prompt length correlates with spread, finding a negligible correlation between the two when jointly considering all models: correlation was $r=0.228$ ($p=1.4\times 10^{-7}$) for mean prompt length when using exact matching prefix as accuracy metric, and $r=-0.022$ ($p=0.615$) for option ranking metric See “On factors influencing spread besides prompt formatting.” in Appendix B.2 for details.

We also added a Limitations section (Appendix C), thoroughly describing the limitations of our methodology.

Finally, we fixed a typo in Section 4.4: we use LLaMA’s hidden representations, not its logits, as the footnote used to say.

Once again, we thank reviewers for their feedback and we're happy to answer any additional questions or discuss remaining concerns!

---

### Meta-Review · Area_Chair_Rq9m · 2023-12-12

**Metareview:**

The design of prompts is a crucial factor in the performance of large language models (LLMs). This paper examines the sensitivity of LLMs to prompt formatting choices in few-shot settings, demonstrating that even minor changes in prompt formatting can have a significant impact on model behavior. The authors find that this sensitivity persists even when the model size or the number of few-shot examples is increased.
To address this issue, the authors propose an algorithm called FORMATSPREAD, which evaluates a set of sampled prompt formats for a given task. FORMATSPREAD efficiently searches the space of plausible prompt formats within a specified computational budget, making it a useful tool for optimizing LLM performance.

Overall, the problem studied in this paper is highly relevant and important for the robust application of LLMs. The paper is well-written and provides valuable insights into the role of prompt formatting in LLM performance.

**Justification For Why Not Higher Score:**

N/A

**Justification For Why Not Lower Score:**

The problem studied in this paper is highly relevant and important for the robust application of LLMs. The paper is well-written and provides valuable insights into the role of prompt formatting in LLM performance.

---

### Decision · Program_Chairs · 2024-01-16

Accept (poster)